# IN THE ZONE: MEASURING DIFFICULTY AND PROGRESSION IN CURRICULUM GENERATION

## ABSTRACT

A common strategy in curriculum generation for reinforcement learning is to train a teacher network to generate tasks that enable student learning. But, what kind of tasks enables this? One answer is tasks belonging to a student's zone of proximal development (ZPD), a concept from developmental psychology. These are tasks that are not too easy and not too hard for the student. Albeit intuitive, ZPD is not well understood computationally. We propose ZONE, a novel computational framework that operationalizes ZPD. It formalizes ZPD through the language of Bayesian probability theory, revealing that tasks should be selected by difficulty (the student's probability of task success) and learning progression (the degree of change in the student's model parameters). ZONE instantiates two techniques that enforce the teacher to pick tasks within the student's ZPD. One is REJECT , which rejects tasks outside of a difficulty scope, and the other is GRAD, which prioritizes tasks that maximize the student's gradient norm. We apply these techniques to existing curriculum learning algorithms. We show that they improve the student's generalization performance on discrete MiniGrid environments and continuous control MuJoCo domains with up to $9\times$ higher success. ZONE also accelerates the student's learning by training with $10\times$ less data.

## 1 INTRODUCTION

Many reinforcement learning (RL) problems require designing a distribution of tasks to train effective policies for the real-world (Taylor & Stone, 2009). However, designing the full space of tasks is challenging; the real world is complicated and specifying every edge task is impractical or even infeasible for certain domains (Wang et al., 2019; Parker-Holder et al., 2022b). These tasks might also be onerous for the agent to solve without the provision of scaffolding. Training the RL agent on all possible tasks is intractable under a limited training budget (Schmidhuber, 2013; Narvekar; Zeng et al., 2022; Florensa et al., 2018a).

The state-of-the-art approaches to this problem use multi-agent curriculum generation algorithms for automating task generation. A *teacher* agent learns to generate tasks judiciously to train a *student* agent (Dennis et al., 2020; Du et al., 2022; Campero et al., 2020; Florensa et al., 2018a; Portelas et al., 2020; Matiisen et al., 2020). The teacher is rewarded by the difficulty of tasks it generates. Even though prior methods share intuitions on the teacher objective, there is no framework for understanding the kind of tasks that best enables student learning and how the teacher should be rewarded to this end. These challenges suggest that we should re-assess how we formalize and operationalize the objective for the teacher.

Thus, our work is interested in the following question: *What kind of tasks should the teacher generate and how should the teacher be rewarded?* From the lens of developmental psychology, an answer is that the teacher should be incentivized to generate tasks within the student's zone of proximal development (ZPD) (Vygotsky & Cole, 1978; Vygotsky, 2012; Cole et al., 2005; Shabani et al., 2010). These tasks have two properties: They are within the student's difficulty and accelerate the student's learning progression. Albeit intuitive and widely known, ZPD lacks a computational framework. This makes operationalizing ZPD in the teaching setting difficult.

Our work proposes ZONE as a computational framework that operationalizes ZPD. ZONE isolates the two properties of ZPD tasks into techniques for enforcing the teacher to generate within a student's ZPD. The first technique is REJECT , which omits training on tasks that fall outside the student's

current ability zone—this is akin to rejection sampling. The second technique is GRAD, which rewards the teacher for generating tasks that maximize the norm of the student network's gradient. These are tasks which induce the largest changes in the student's current model. We apply these techniques to two popular curriculum generation algorithms—PAIRED (Dennis et al., 2020) and Goal GAN (Florensa et al., 2018b)—and show that ZONE accelerates student learning on a suite of discrete and continuous environments. We then investigate how these two techniques impact the student's and teacher's learning.

In summary, our work's contributions are the following:

1. We propose ZONE, a novel computational framework that formalizes the zone of proximal development (ZPD) (Vygotsky & Cole, 1978) with Bayesian probability theory.

2. ZONE operationalizes ZPD with two techniques: REJECT , which rejects tasks that fall outside the student's difficulty range, and GRAD, which rewards the teacher for generating tasks that maximize the student's gradient norms.

3. We show that REJECT and GRAD improve the student's generalization performance and learning speed across a variety of discrete and continuous environments.

4. We investigate how the ZONE techniques improve the teacher's ability to generate ZPD tasks for the student.

## 2 RELATED WORKS

**Zone of proximal development (ZPD)** The objective of a teacher is to help a student learn effectively and to maximize the student's long-term performance. Prior work in developmental psychology and personalized education argue that a teacher should aid students in problems that fall within their zone of proximal development (ZPD) (Vygotsky & Cole, 1978). ZPD problems are ones that are not too easy (where the student doesn't need the teacher's assistance), and ones that are not too difficult (where the student couldn't solve even with the teacher's assistance). Aiding students on ZPD problems is how students would most benefit from the teacher's scaffolding, such as through a curriculum (Warford, 2011; Wass & Golding, 2014). Vygotsky's idea has shaped how we think about teaching human students in different domains, like teaching the sciences (Chounta et al., 2017; Vainas et al., 2019) or foreign languages (Mu et al., 2021). It has also shaped our understanding of how young infants acquire knowledge over time (Diaz et al., 1991; Wass & Golding, 2014).

**Teacher-student curriculum generation** Teacher-student curriculum generation is a long-standing paradigm for accelerating training and generalization of RL agents (Matiisen et al., 2020; Sukhbaatar et al., 2018; Florensa et al., 2017; 2018b; Zhang et al., 2020; Parker-Holder et al., 2022b; Dennis et al., 2020; Jiang et al., 2021b; Portelas et al., 2020; Fang et al., 2020; Soviany et al., 2022). These algorithms reward the teacher based on measures of difficulty, such as the student's return. Despite this similarity, there is little work on formalizing what make up good task generations and how to align the teacher's objective to this end. This is contrary to work in developmental psychology that suggest pedagogical interventions via the teacher based on the student's needs and progress.

Our work focuses on two popular curriculum generation algorithms: PAIRED (Dennis et al., 2020) and Goal GAN (Florensa et al., 2018b). PAIRED is a successful regret-based algorithm applied to discrete domains, where an adversarial teacher generates tasks that maximize regret between a student and a competing student (the anti-student). Goal GAN is a similarly successful algorithm designed for continuous control domains where a generative adversarial network (GAN) (Goodfellow et al., 2020) teacher generates 2D navigation goals. We focus on these algorithms because several works build on them as either a competitive baseline or the basis of their own algorithms (Jiang et al., 2021a; Gur et al., 2021; Du et al., 2022; Parker-Holder et al., 2022a; Zhang et al., 2020). Additionally, the algorithms cover different properties of prior work, such as domains (discrete vs. continuous), difficulty criteria (static vs. dynamic), and teacher objectives (regret vs. non-regret). Prior intrinsic reward methods can alternatively enable generalization where additional reward supplements the extrinsic reward to incentivize exploration of the environment (Campero et al., 2020; Zhang et al., 2021; Raileanu & Rocktäschel, 2019). However, our work focuses on the teacher determining the student's curriculum and considers these methods out of scope.

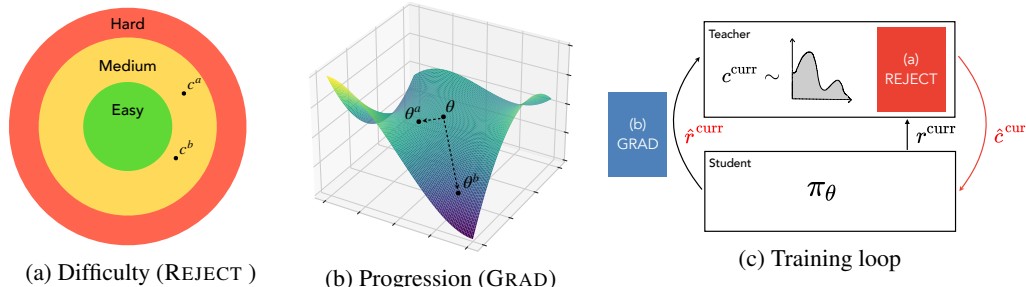

(a) Difficulty (REJECT )    (b) Progression (GRAD)    (c) Training loop

Figure 1: ZONE consists of two components ensuring the teacher generates tasks within the student's zone of proximal development. (a) is REJECT , where tasks outside of a difficulty range are omitted from training. Tasks that are too hard (in red) and too easy (in green) are rejected. It keeps tasks $c^a, c^b$. (b) is GRAD where the teacher selects tasks that induce large changes in the student's model. The plot is an illustrative example of the model space in $\mathbb{R}^3$. The student's current model is $\theta$. The student's model after updating on task $c^a$ is $\theta^a$ and after updating on task $c^b$ is $\theta^b$. $c^b$ induces a larger change than $c^a$, thus the teacher prefers $c^b$. (c) situates both ZONE components in a typical teacher-student training loop. REJECT  rejects tasks $c^{\text{curr}}$ based on the student's reward $r^{\text{curr}}$, and GRAD encourages the teacher to generate tasks that maximize the student's gradient norm.

**ZPD-based teaching**   A teacher acting on ZPD generates tasks with two types of properties. The first property focuses on difficulty: curriculum tasks are neither too easy nor too hard. In prior work like PAIRED (Dennis et al., 2020) and Goal GAN (Florensa et al., 2018b), a teacher generates a batch of tasks and trains the student on the entire batch regardless of whether the tasks meet the difficulty criterion. Training the student on problems that are too easy and too hard slows down the student's learning. In contrast, our work proposes a sample-efficient rejection sampling technique inspired by ZONE: we replace tasks that don't meet the difficulty criterion with those that do, and train on the resulting batch.

The second property focuses on progression: curriculum tasks accelerate the student's ability to generalize. In general, identifying such tasks prior to training the student is difficult. A proxy for identifying such tasks is picking those that induce large model changes in the student, such as the norm of the student's model gradients. Prior work in the supervised learning setting have employed the gradient norm as a heuristic in active learning (Ash et al., 2019; Jiang et al., 2019). Prior work in intrinsic rewards for RL agents has used the gradient norm for learning better world models (Sun et al., 2011; Houthooft et al., 2016). However, to the best of our knowledge, the idea of maximizing the gradient norm of a student has not been used as a curriculum learning objective.

## 3   METHOD

The goal of a teacher is to generate tasks that lead to the student's success in the long-run while also accelerating the pace of student learning. This is a challenging goal to convert into an objective: How does a teacher measure both the long-run performance and progression in selecting the best tasks? Our work is interested in mathematically formalizing the teacher's objective motivated by ZPD. The formalism motivates the techniques we use to improve existing curriculum generation algorithms. We first describe the teacher-student's learning setting. Then, we formalize ZONE, a framework for describing the ideal teacher objective with the language of Bayesian probability theory. Finally, we describe how ZONE can improve existing curriculum generation algorithms.

### 3.1   TEACHER-STUDENT FORMALISM

Our work is concerned with agents that endeavor to learn optimal behaviors across multiple tasks (Kaelbling, 1993; Schaul et al., 2015), a problem we formulate as a Contextual MDP (Brunskill & Li, 2013; Hallak et al., 2015; Modi et al., 2018) (CMDP) given by $\mathcal{M} = \langle \mathcal{S}, \mathcal{A}, \mathcal{C}, \mathcal{R}, \mathcal{T}, \mu, \gamma, \chi \rangle$. Here, each task of interest is identified by an individual context contained in $\mathcal{C}$ while jointly sharing the same set of states $\mathcal{S}$, set of actions $\mathcal{A}$, and discount factor $\gamma \in [0, 1)$. We will refer to contexts and tasks interchangeably. Each variation in the context induces a new infinite-horizon, discounted Markov Decision Process (MDP) (Bellman, 1957; Puterman, 1994) that may differ in the initial state distribution $\mu : \mathcal{C} \rightarrow \Delta(\mathcal{S})$, the (deterministic) reward function $\mathcal{R} : \mathcal{C} \times \mathcal{S} \times \mathcal{A} \rightarrow [0, 1]$ providing evaluative feedback signals (in the unit interval), or the transition function $\mathcal{T} : \mathcal{C} \times \mathcal{S} \times \mathcal{A} \rightarrow \Delta(\mathcal{S})$ prescribing distributions over next states. $\chi \in \Delta(\mathcal{C})$ denotes a distribution over contexts, ie. a task distribution.

The student is a policy parameterized by $\theta \in \Theta \subseteq \mathbb{R}^d$ and is a mapping to distributions over actions given a state $\pi_\theta : \mathcal{S} \to \Delta(\mathcal{A})$. The teacher is a distribution over contexts parameterized by $\phi$, $\chi_\phi \in \Delta(\mathcal{C})$. At the beginning of an episode, the teacher samples a context $c \sim \chi_\phi(\cdot)$ which is held fixed as the student contends with learning optimal behavior in the resulting MDP $\langle \mathcal{S}, \mathcal{A}, \mathcal{R}_c, \mathcal{T}_c, \mu_c, \gamma \rangle$. At each discrete timestep $t \in \mathbb{N}$, beginning with an initial state $s_0 \sim \mu_c(\cdot)$, the student observes the current state $s_t \in \mathcal{S}$, executes an action $a_t \in \mathcal{A}$, observes a reward $r_t = \mathcal{R}_c(s_t, a_t)$, and transitions to the next state $s_{t+1} \sim \mathcal{T}_c(\cdot \mid s_t, a_t)$.

## 3.2 Formalizing the Zone: How should the teacher be rewarded?

To help build intuition for ZONE we describe a one-step curriculum learning process using the language of Bayesian probability theory. The goal of the teacher is to learn to construct a curriculum over tasks $\mathcal{C}$ such that the student does well in the long run. Because we care about generalization, we model the student's long-run performance as its test success rate. $R$ is a Bernoulli random variable that denotes success: If the student solves the task, $R = 1$, otherwise $R = 0$ (Levine, 2018)[1]. We use $C^{\text{curr}}, R^{\text{curr}}$ to denote the curriculum task and student's curriculum success, $C^{\text{test}}, R^{\text{test}}$ denote the test task and student's test success. Similarly, $\theta^{\text{curr}} \in \mathbb{R}^d$ denotes the student's policy parameters at the beginning of the curriculum. The teacher's objective is learn task distribution $\chi_\phi(C^{\text{curr}})$ to maximize the student's expected success over a held-out task distribution:

$$\max_{\chi_\phi(C^{\text{curr}})} \mathbb{E}_{c^{\text{curr}} \sim \chi_\phi(\cdot)}[p(R^{\text{test}} = 1 | C^{\text{test}}, C^{\text{curr}} = c^{\text{curr}}, \theta^{\text{curr}})] \tag{1}$$

The objective does not yet reflect the student's learning update after training on $C^{\text{curr}}$. To do so, we introduce the random variable $R^{\text{curr}}$ to model the student's curriculum success. We also introduce $\theta^{\text{test}}$, which is the student's policy parameters *after* updating. We apply the chain rule to expand Equation 1 as follows:

$$\max_{\chi_\phi(C^{\text{curr}})} \mathbb{E}_{c^{\text{curr}} \sim \chi_\phi(\cdot)}[p(R^{\text{test}} = 1 | C^{\text{test}}, c^{\text{curr}}, \theta^{\text{curr}})] \tag{2}$$

$$= \max_{\chi_\phi(C^{\text{curr}})} \mathbb{E}_{c^{\text{curr}} \sim \chi_\phi(\cdot)}\left[ \sum_{r^{\text{curr}} \in R} \int_{\mathbb{R}^d} p(R^{\text{test}} = 1, r^{\text{curr}}, \theta^{\text{test}} | C^{\text{test}}, c^{\text{curr}}, \theta^{\text{curr}}) d\theta^{\text{test}} \right] \tag{3}$$

$$= \max_{\chi_\phi(C^{\text{curr}})} \mathbb{E}_{c^{\text{curr}} \sim \chi_\phi(\cdot)}\left[ \sum_{r^{\text{curr}} \in R} \int_{\mathbb{R}^d} p(R^{\text{test}} = 1 | C^{\text{test}}, \theta^{\text{test}}) p(\theta^{\text{test}} | r^{\text{curr}}, c^{\text{curr}}, \theta^{\text{curr}}) p(r^{\text{curr}} | c^{\text{curr}}, \theta^{\text{curr}}) d\theta^{\text{test}} \right] \tag{4}$$

$$\geq \max_{\chi_\phi(C^{\text{curr}})} \mathbb{E}_{c^{\text{curr}} \sim \chi_\phi(\cdot)}\left[ \sum_{r^{\text{curr}} \in R} \int_{\theta^* \in \Theta^*} p(R^{\text{test}} = 1 | C^{\text{test}}, \theta^*) p(\theta^* | r^{\text{curr}}, c^{\text{curr}}, \theta^{\text{curr}}) p(r^{\text{curr}} | c^{\text{curr}}, \theta^{\text{curr}}) d\theta^* \right] \tag{5}$$

The transition from Equation 3 to Equation 4 uses conditional independence: The student's test-time performance $R^{\text{test}}$ conditioned on its updated model parameters $\theta^{\text{test}}$ is independent of $C^{\text{curr}}, R^{\text{curr}}$. Moving from Equation 4 to Inequality 5 assumes that the teacher only cares about the student learning *optimal* test parameters $\theta^*$. $\Theta^*$ is the set of optimal test parameters that globally maximize the student's test performance $p(R^{\text{test}} = 1 | C^{\text{test}}, \theta^*)$. We assume the teacher only cares about maximizing the student's test performance because the teacher's objective is to help the student perform successfully in the long run. We show in Appendix A that we can lower-bound Equation 4 with 5.

What does the decomposition in the final equation tell us about the teacher's role in the student's long-run performance? The teacher influences the difficulty of the student's curriculum $p(r^{\text{curr}} | c^{\text{curr}}, \theta^{\text{curr}})$. The teacher also influences the student's learning progression to good test-time parameters, $p(\theta^* | r^{\text{curr}}, c^{\text{curr}}, \theta^{\text{curr}})$. It finds tasks that accelerate the student's acquisition of optimal

---

[1]In Appendix E, we empirically show that the choice of difficulty measure matters. For example, one cannot simply use the environment's dense reward function. The dense reward values are not always correlated with task success.

---

**Algorithm 1** ZONE (**REJECT** and **GRAD**) applied to a standard curriculum generation loop.

---

Initialize student $\pi_\theta$, teacher $\chi_\phi$, and curriculum learning algorithm $alg$
**while** not converged **do**
$\quad \{c_i^{\text{curr}}\}_{i=1}^K \sim \chi_\phi(C)$ $\qquad\qquad\qquad\qquad\qquad\qquad\qquad$ ▷ generate tasks
$\quad \{r_i^{\text{curr}}\}_{i=1}^K \leftarrow$ rollout$(\pi_\theta, \{c_i^{\text{curr}}\}_{i=1}^K)$ $\qquad\qquad\qquad\qquad$ ▷ evaluate tasks
$\quad \{\hat{c}_i^{\text{curr}}\}_{i=1}^K, \{\hat{r}_i^{\text{curr}}\}_{i=1}^K \leftarrow alg.$criterion $(\{c_i^{\text{curr}}\}_{i=1}^K, \{r_i^{\text{curr}}\}_{i=1}^K)$ $\qquad$ ▷ **REJECT**
$\quad \pi_\theta, \Delta\theta \leftarrow$ train_student$(\pi_\theta, \{\hat{c}_i^{\text{curr}}\}_{i=1}^K, \{\hat{r}_i^{\text{curr}}\}_{i=1}^K)$ $\qquad\qquad$ ▷ train student
$\quad \chi_\phi \leftarrow$ train_teacher$(\chi_\phi, \{\hat{c}_i^{\text{curr}}\}_{i=1}^K, \{\hat{r}_i^{\text{curr}}\}_{i=1}^K, \Delta\theta)$ $\qquad$ ▷ train teacher with **GRAD**
**end while**

---

test parameters. ZONE elucidates why the teacher might want to avoid tasks that are too easy and too hard. Easy tasks (high $p(R^{\text{curr}} = 1 | c^{\text{curr}}, \theta^{\text{curr}})$) do not change the student's current model; if the student does not have a good model ($\theta^{\text{curr}} \neq \theta^*$), then picking an easy task will not increase the chances of the student learning a better test model (low $p(\theta^* | r^{\text{curr}}, c^{\text{curr}}, \theta^{\text{curr}})$). The teacher should avoid picking hard tasks too: the probability of failure is high, and the student will likely need prolonged interaction with difficult task instances before learning a good test model. In Appendix B, we provide a more formal reinforcement learning analysis that recovers a similar decomposition to the above for the ZONE framework.

### 3.3 USING ZONE TO IMPROVE CURRICULUM ALGORITHMS

In general, it is hard to exactly estimate $p(r^{\text{curr}} | c^{\text{curr}}, \theta^{\text{curr}})$ and $p(\theta^* | r^{\text{curr}}, c^{\text{curr}}, \theta^{\text{curr}})$ per task instance $c^{\text{curr}}$. Below, we explain two ZONE techniques to make the formalism practical. These techniques are illustrated in Figure 1 and Algorithm 1 as simple additions to the teacher-student training loop.

**Modeling $p(r^{\text{curr}} | c^{\text{curr}}, \theta^{\text{curr}})$ with rejection sampling (REJECT )** Prior works including Goal GAN and PAIRED measure difficulty as the core objective. They typically approximate problem difficulty with the student's return. Broadly speaking, the algorithms reward the teacher for generating tasks where the student achieves a cumulative reward $r \in [r_{\min}, r_{\max}]$. In other words, the teacher satisfies the algorithm's difficulty criterion if the teacher does not generate problems that are too easy ($r > r_{\max}$) and too hard ($r < r_{\min}$). This range may change (as in PAIRED) or be fixed (as in Goal GAN) throughout training. Rejecting tasks that fall outside the range ensures that the teacher generates tasks within a desired level of difficulty. We model REJECT as $p(r^{\text{curr}} | c^{\text{curr}}, \theta^{\text{curr}})$ because the teacher influences the difficulty of the curriculum and task MDP. The procedure for REJECT is as follows: At every round, the teacher generates a batch of $K$ tasks $\{c_i^{\text{curr}}\}_{i=1}^K$. The student runs on the tasks and achieves the cumulative reward of $\{r_i^{\text{curr}}\}_{i=1}^K$. The rewards are checked against the algorithm's criterion; if the reward does not satisfy the criterion, the task is removed from the batch. Otherwise, it remains in the batch. Rather than let the teacher waste generated tasks and use a smaller batch, we upsample the accepted tasks to replace the removed tasks[2]. The student is trained on the resulting batch $\{\hat{c}_i^{\text{curr}}\}_{i=1}^K$. Prior works like Wang et al. (2019) and Brant & Stanley (2017) use rejection as a heuristic to improve evolutionary algorithms. Nonetheless, our work formalizes REJECT with the ZONE analysis and shows that REJECT can be broadly useful for existing curriculum learning algorithms in the literature.

**Modeling $p(\theta^* | r^{\text{curr}}, c^{\text{curr}}, \theta^{\text{curr}})$ with maximizing gradient norms (GRAD)** In maximizing the lower bound in Inequality 5, the teacher wants to generate tasks which move the student's parameters to a global maximizer $\theta^*$. On the one hand, knowing $\theta^*$ or the set of global maximizers $\Theta^*$ a priori is a challenge. On the other hand, updating towards $\theta^*$ is not possible unless the student's model is moving. Thus, we propose a simple proxy that incentivizes the student's model to change. Namely, we reward the teacher for generating tasks that induce high model gradient norms in the student. To the best of our knowledge, the idea of maximizing the gradient norm of a student has not been explored as a curriculum learning objective. Both Goal GAN and PAIRED use actor-critic algorithms, therefore the gradient norm is the norm of the actor and the critic gradients. In practice, GRAD calculates the norm of the student's gradient and adds it to the teacher's original

---

[2]We are not running the teacher or the student more often when we do rejection sampling. The computational requirements are the same with or without ZONE.

reward from prior works: $r_{\text{teacher}} = r_{\text{original}} + \|\Delta\theta\|_2$. The teacher's reward can also be casted as a weighted sum of the original reward and the student's gradient norm; our work does not explore the importance of tuning these weights. For dense reward settings, GRAD multiplies the two terms: $r_{\text{teacher}} = r_{\text{original}} \cdot \|\Delta\theta\|_2$. In dense reward settings, we found that $r_{\text{original}}$ was often negative and much smaller in magnitude than the gradient norm. If we add the two terms, the teacher's reward would be dominated by the gradient norm.

## 4   EXPERIMENTS

We now evaluate the ability of ZONE to inform and improve existing curriculum generation algorithms. Specifically, we aim to answer the following research questions (RQs):

**RQ1:** *How does ZONE impact the student's generalization performance and its learning progress?* Section 5.1 investigates impact of ZONE on the student's learning by applying ZONE on the curriculum generation algorithms on a suite of discrete and continuous environments. We compare the held-out performance of the students with and without ZONE.

**RQ2:** *How does ZONE impact the teacher's learning and task generations?* Section 5.2 investigates this question by qualitatively and quantitatively comparing the task generations from ZONE teachers and non-ZONE teachers. We also compare the teachers on metrics that capture whether the teachers generate ZPD tasks and whether they learn to generate ZPD tasks more easily.

**Algorithms and applications of ZONE**   As previously mentioned, we run ZONE techniques over two existing curriculum generation algorithms Goal GAN (Florensa et al., 2018b) and PAIRED (Dennis et al., 2020). We use each algorithm's default model settings and hyperparameters. Goal GAN's difficulty criterion is that the student achieves returns of $r \in [0.1, 0.9]$ on the generated goals. This is the interval chosen by prior work to avoid problems that are too easy and too hard. The teacher is trained on labelled goals of whether the goals are within the difficulty range. Goal GAN+REJECT trains on tasks that are *only* within that difficulty range. Because the teacher is not trained with rewards, we do not apply GRAD to Goal GAN. PAIRED trains the teacher to maximize regret $r_{\text{teacher}} = \max(r_{-s} - r_s, 0)$, where $r_s$ is the reward of the student and $r_{-s}$ is the reward of the competing student. As regret must be positive (Bell, 1982; Loomes & Sugden, 1982; Lattimore & Szepesvári, 2020), PAIRED's difficulty criterion is that the student achieves returns where $r_{-s} - r_s > 0$ on the generated tasks. In other words, PAIRED+REJECT trains the student on tasks where $r_s \in [0, r_{-s}]$. PAIRED+GRAD trains the student on all tasks, but rewards the teacher with $r_{\text{teacher}} = \max(r_{-s} - r_s, 0) + \|\Delta\theta\|$ on sparse reward settings and $r_{\text{teacher}} = \max(r_{-s} - r_s, 0) \cdot \|\Delta\theta\|$ on dense reward settings. The student's gradient norm include both the gradients of the student's actor and critic networks. PAIRED+REJECT +GRAD trains the student on the tasks within the difficulty range, and rewards the teacher as in PAIRED+GRAD. More details on these algorithms can be found in the Appendix D.

**Environments**   We run each algorithm on its original domains. PAIRED's original domains are shown in Figure 2a,2b,2c,2d which are partially observable navigation tasks in the MiniGrid environment (Chevalier-Boisvert et al., 2018). Goal GAN's original domains are shown in Figure 2e,2f,2g). We also run both algorithms on additional MuJoCo continuous domains where students are trained within an in-domain region, and tested on a held-out region (Figure 2h,2i,2k2j). We use Du et al. (2022)'s implementation for this setting. Note that in their implementation, the student is trained using *dense* rewards whereas the original algorithms are trained on sparse reward settings. Nonetheless, the teacher still rejects tasks based on the student's (binary) success, regardless of the exact numerical return achieved.

## 5   RESULTS

### 5.1   HOW DOES ZONE IMPACT THE STUDENT'S GENERALIZATION AND PROGRESS?

To test whether ZONE improves prior generation algorithms, we apply the ZONE techniques to PAIRED and Goal GAN and measure the student's test-time performance. We mark the techniques

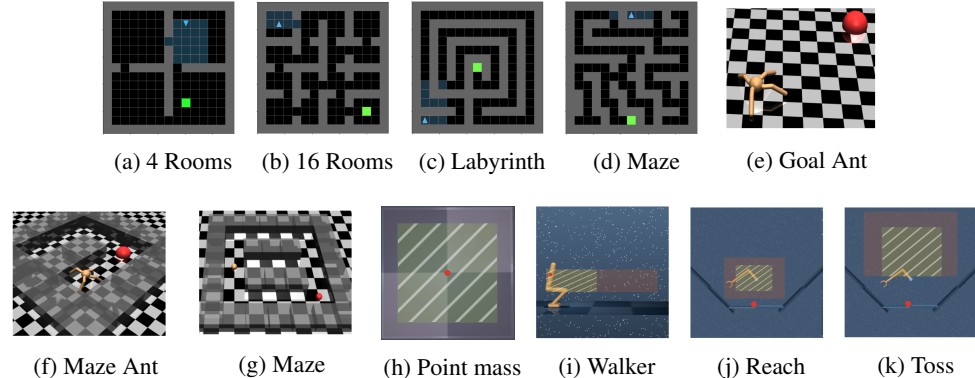

| (a) 4 Rooms | (b) 16 Rooms | (c) Labyrinth | (d) Maze | (e) Goal Ant |

| (f) Maze Ant | (g) Maze | (h) Point mass | (i) Walker | (j) Reach | (k) Toss |

Figure 2: PAIRED transfer environments (a-d) (Dennis et al., 2020), Goal GAN environments (e-g) (Florensa et al., 2018a), and additional MuJoCo environments adapted from Du et al. (2022) (h-k). In the MuJoCo environments, the green-shaded area corresponds to the space of in-domain goals the teacher can generate, and the gray-shaded area corresponds to the out-of-domain goals the student is tested on.

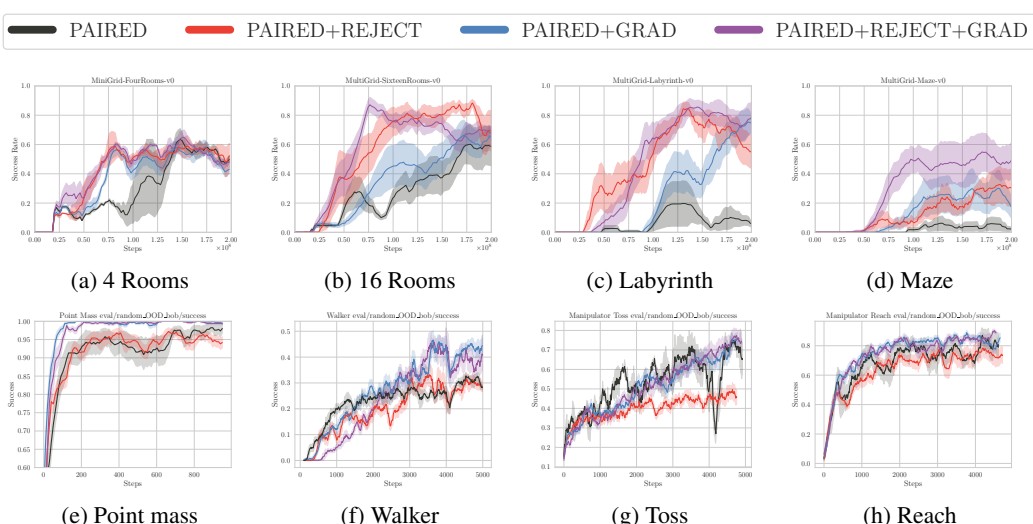

| (a) 4 Rooms | (b) 16 Rooms | (c) Labyrinth | (d) Maze |

| (e) Point mass | (f) Walker | (g) Toss | (h) Reach |

Figure 3: PAIRED performance on MiniGrid transfer environments (a-d) and MuJoCo control environments (e-h). All the methods see the same amount of data per step; REJECT 's dropped samples are included in the learning curves.

REJECT in **red**, GRAD in **blue**, REJECT +GRAD in **purple**. The original algorithms are marked in **black** in the graphs. We test PAIRED on its original transfer environments and MuJoCo environments (Figure 3), and test Goal GAN on its original test environments and MuJoCo environments (Figure 4).

The ZONE techniques propose tasks that improve and accelerate the student's performance on a majority of the domains. In the MiniGrid setting (top row of Figure 3), PAIRED+REJECT discards problems that inhibit student learning in regular PAIRED. PAIRED+GRAD is slower to find tasks that help the student generalizes, however matches REJECT 's performance over time (eg. in Sixteen Rooms, Labryinth and Maze). On all four domains, PAIRED+REJECT +GRAD performs at least as well as both individual techniques. This is most evident on Maze and demonstrates that REJECT and GRAD interact positively with each other.

In the control environments like the Goal GAN environments or MuJoCo environments (bottom row of Figure 3 and Figure 4), ZONE techniques provide slight improvements over the original algorithm. Goal GAN generally improves with REJECT , most noticeably on Walker, Maze, and Maze Ant in

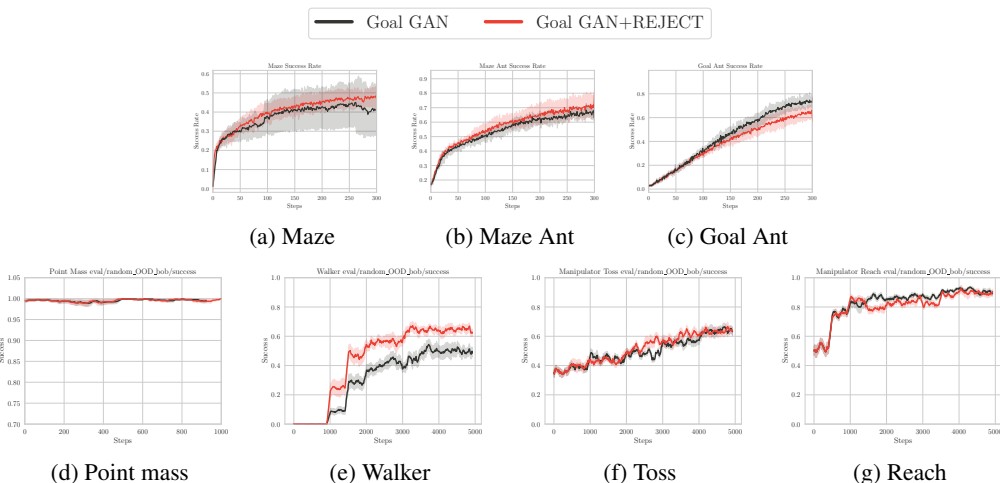

Figure 4: Goal GAN performance on Goal GAN environments (a-c), and on the MuJoCo control environments (d-g).

Figure 4. In Figure 3, on the dense reward MuJoCo environments, the student benefits from GRAD across all domains, however does not benefit from REJECT . We hypothesize REJECT is *especially* helpful in the sparse reward setting as that is where learning a good policy is most challenging. REJECT is not helpful in a dense reward setting like the MuJoCo environments because the agent learns on every trajectory, or in simple navigation environments like the Goal GAN environments.

Altogether, these results suggest that ZONE aids student' learning, particularly when the task structure and the teacher's task generations are complex, such as in the MiniGrid environment. In the MiniGrid setting, the teacher generates a task that is a $15 \times 15$ grid with up to $50$ walls. Small variations in its generations can result in large variations of the task difficulty. For example, placing a single wall piece at the end of a corridor might block off a direct route to the goal for the student. By contrast, in the control environments, small variations the 2D goals would not make a large difference in the task difficulty.

## 5.2   HOW DOES ZONE IMPACT THE TEACHER'S TASK GENERATIONS AND LEARNING?

This section investigates how the teacher is impacted by ZONE by comparing the tasks generated by the base algorithm and the ZONE variant. We focus on the MiniGrid setting because we see the largest improvements in this setting, and particularly compare between PAIRED and PAIRED+REJECT +GRAD. We include the analysis on the other algorithms and environments in Appendix C. The section focuses on two items. The first is an analysis of the task generations. The second is an analysis of the teacher quality; we use the rejection rate and the student's gradient norms as proxy measures for this.

**Analysis of task generations**   We first compare the qualitative differences in generated tasks. Table 1 shows a series of tasks generated by PAIRED (top row) and PAIRED+REJECT +GRAD (bottom row). The columns indicate number of updates to the teacher over time. PAIRED generates simpler tasks with fewer obstacles than PAIRED+REJECT +GRAD. This is most noticeable within the first 8,000 updates. We confirm this in Figure 5a where we measure the percentage of area covered in obstacles within the first $10\%$ of student training time; Figure 5a shows that PAIRED+REJECT +GRAD generates more obstacle-filled tasks than PAIRED. Furthermore, columns 12-24k indicate that PAIRED+REJECT +GRAD generates tasks that contain more hallways, eg. along the bottom of the grid. These complex tasks are helpful for the student in generalizing to human-designed transfer environments, like a labyrinth or a maze. Note that while these tasks are more complex, they are also feasible since we apply REJECT . Altogether, this provides evidence that ZONE techniques are effective in helping the teacher produce a coverage of difficult and interesting grid worlds that improve the student's generalization.

| | 4k | 8k | 12k | 16k | 20k | 24k |
|---|---|---|---|---|---|---|

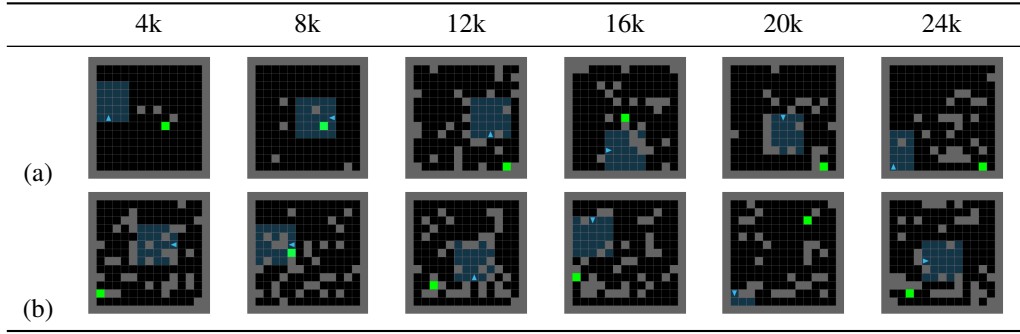

(a)

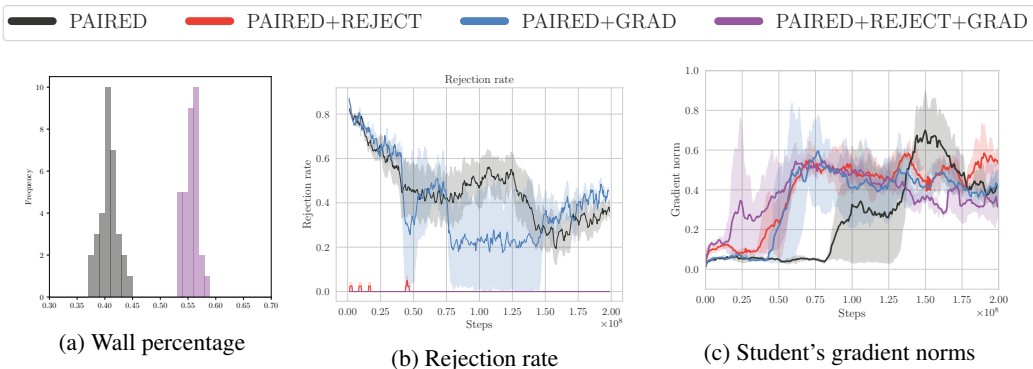

(b)

Table 1: (a) row are tasks generated from PAIRED, and (b) row are tasks generated from PAIRED+REJECT+GRAD. The columns mark the number of model updates.

| ━━ PAIRED | ━━ PAIRED+REJECT | ━━ PAIRED+GRAD | ━━ PAIRED+REJECT+GRAD |

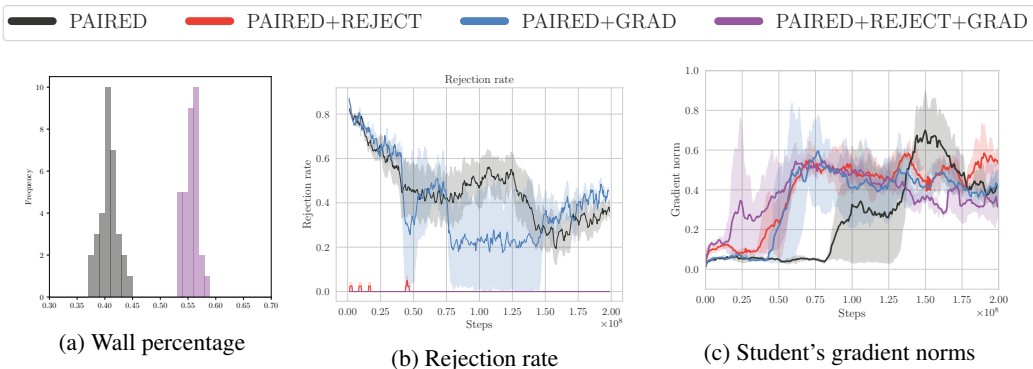

(a) Wall percentage    (b) Rejection rate    (c) Student's gradient norms

Figure 5: (a) tracks the percentage of grid space occupied by walls, generated by the PAIRED teacher and the PAIRED+REJECT +GRAD. (b) tracks the rejection rate of the teacher based on the PAIRED difficulty criterion. (c) tracks the student's gradient norms over the course of training.

**Analysis on teacher quality** We now investigate whether the teacher is able to satisfy the ZPD criteria of difficulty and progression. Do the ZONE techniques help the teacher generate tasks that meet the difficulty criterion *and* that maximize the student's gradient norm? Figure 5b compares the rejection rates across the teachers. The rejection rate is calculated as the percentage of tasks that do not satisfy PAIRED's regret objective in a sample batch. Rejection in PAIRED occurs when the student does better than or equal to the anti-student. All ZONE variants learn to generate tasks satisfying PAIRED's difficulty criterion much faster than the original PAIRED algorithm. PAIRED+REJECT and PAIRED+REJECT +GRAD are the fastest to do so. Figure 5c compares the student's gradient norms on the teacher's tasks. The ZONE techniques are very useful in guiding the teacher to generate tasks which induce higher gradient norms than PAIRED: PAIRED+REJECT results in higher gradient norms than regular PAIRED, and PAIRED+REJECT +GRAD results in higher gradient norms than both PAIRED+REJECT and PAIRED+GRAD. This demonstrates that REJECT helps select examples that lead to high gradient norms for the student. This also suggests that the measure of progression could still be improvement. Overall, these results indicate that ZONE aids the teacher in learning to generate tasks better which are within the student's difficulty range and lead to faster learning.

## 6 CONCLUSION

Our work is interested in the kind of tasks that best lead to student learning and the teacher's objective. We take inspiration from the concept of the "zone of proximal development" (ZPD) in developmental psychology, which asserts that students learn most effectively when given tasks that are not too difficult and accelerate their learning. We propose ZONE as a computational framework that operationalizes ZPD. ZONE consists of two simple techniques: rejecting tasks that fall outside of a difficulty zone (REJECT ) and maximizing the student's gradient norms (GRAD). We show that these techniques improve the student's performance on both continuous and discrete domains. Altogether, this work proposes simple techniques that could be useful for practitioners who want to both accelerate the speed of RL training and improve the agent's generalization capabilities. The ZONE methods might be most effective in settings with complex task structures (eg. in a natural language setting for reasoning domains) as the results in MiniGrid suggest.

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

## A  LOWER BOUNDING THE TEACHER OBJECTIVE

We are interested in showing that the lower bound used in Section 3.2 holds. As a reminder, the lower bound is:

$$
\max_{\chi_\phi(C^{\text{curr}})} \mathbb{E}_{c^{\text{curr}} \sim \chi_\phi(\cdot)} \left[ \sum_{r^{\text{curr}} \in R} \int_{\mathbb{R}^d} p(R^{\text{test}} = 1 | C^{\text{test}}, \theta^{\text{test}}) p(\theta^{\text{test}} | r^{\text{curr}}, c^{\text{curr}}, \theta^{\text{curr}}) p(r^{\text{curr}} | c^{\text{curr}}, \theta^{\text{curr}}) d\theta^{\text{test}} \right]
$$

$$
\geq \max_{\chi_\phi(C^{\text{curr}})} \mathbb{E}_{c^{\text{curr}} \sim \chi_\phi(\cdot)} \left[ \sum_{r^{\text{curr}} \in R} \int_{\theta^* \in \Theta^*} p(R^{\text{test}} = 1 | C^{\text{test}}, \theta^*) p(\theta^* | r^{\text{curr}}, c^{\text{curr}}, \theta^{\text{curr}}) p(r^{\text{curr}} | c^{\text{curr}}, \theta^{\text{curr}}) d\theta^* \right]
$$

To see how this is true, we do the following:

$$\max_{\chi_\phi(C^{\text{curr}})} \mathbb{E}_{c^{\text{curr}} \sim \chi_\phi(\cdot)} \left[ \sum_{r^{\text{curr}} \in R} \int_{\mathbb{R}^d} p(R^{\text{test}} = 1 | C^{\text{test}}, \theta^{\text{test}}) p(\theta^{\text{test}} | r^{\text{curr}}, c^{\text{curr}}, \theta^{\text{curr}}) p(r^{\text{curr}} | c^{\text{curr}}, \theta^{\text{curr}}) d\theta^{\text{test}} \right] \tag{6}$$

$$= \max_{\chi_\phi(C^{\text{curr}})} \mathbb{E}_{c^{\text{curr}} \sim \chi_\phi(\cdot)} \left[ \sum_{r^{\text{curr}} \in R} \int_{\mathbb{R}^d} p(R^{\text{test}} = 1 | C^{\text{test}}, \theta^{\text{test}}) p(\theta^{\text{test}} | r^{\text{curr}}, c^{\text{curr}}, \theta^{\text{curr}}) \tag{7}$$

$$\mathbb{1} \left[ \theta^{\text{test}} \in \operatorname{supp} p(\theta^{\text{test}} | r^{\text{curr}}, c^{\text{curr}}, \theta^{\text{curr}}) \right] p(r^{\text{curr}} | c^{\text{curr}}, \theta^{\text{curr}}) d\theta^{\text{test}} \right] \tag{8}$$

$$= \max_{\chi_\phi(C^{\text{curr}})} \mathbb{E}_{c^{\text{curr}} \sim \chi_\phi(\cdot)} \left[ \sum_{r^{\text{curr}} \in R} \int_{\mathbb{R}^d} p(R^{\text{test}} = 1 | C^{\text{test}}, \theta^{\text{test}}) p(\theta^{\text{test}} | r^{\text{curr}}, c^{\text{curr}}, \theta^{\text{curr}}) \tag{9}$$

$$\left( \mathbb{1} \left[ \theta^{\text{test}} \in \max_{\theta \text{ s.t. } p(\theta^{\text{test}} | r^{\text{curr}}, c^{\text{curr}}, \theta^{\text{curr}}) > 0} p(R^{\text{test}} = 1 | C^{\text{test}}, \theta^{\text{test}}) \right] \right. \tag{10}$$

$$\left. + \mathbb{1} \left[ \theta^{\text{test}} \notin \max_{\theta \text{ s.t. } p(\theta^{\text{test}} | r^{\text{curr}}, c^{\text{curr}}, \theta^{\text{curr}}) > 0} p(R^{\text{test}} = 1 | C^{\text{test}}, \theta^{\text{test}}) \right] \right) p(r^{\text{curr}} | c^{\text{curr}}, \theta^{\text{curr}}) d\theta^{\text{test}} \right] \tag{11}$$

$$\geq \max_{\chi_\phi(C^{\text{curr}})} \mathbb{E}_{c^{\text{curr}} \sim \chi_\phi(\cdot)} \left[ \sum_{r^{\text{curr}} \in R} \int_{\mathbb{R}^d} p(R^{\text{test}} = 1 | C^{\text{test}}, \theta^{\text{test}}) p(\theta^{\text{test}} | r^{\text{curr}}, c^{\text{curr}}, \theta^{\text{curr}}) \tag{12}$$

$$\mathbb{1} \left[ \theta^{\text{test}} \in \max_{\theta \text{ s.t. } p(\theta^{\text{test}} | r^{\text{curr}}, c^{\text{curr}}, \theta^{\text{curr}}) > 0} p(R^{\text{test}} = 1 | C^{\text{test}}, \theta^{\text{test}}) \right] p(r^{\text{curr}} | c^{\text{curr}}, \theta^{\text{curr}}) d\theta^{\text{test}} \right] \tag{13}$$

$$\geq \max_{\chi_\phi(C^{\text{curr}})} \mathbb{E}_{c^{\text{curr}} \sim \chi_\phi(\cdot)} \left[ \sum_{r^{\text{curr}} \in R} \int_{\theta^* \in \Theta^*} p(R^{\text{test}} = 1 | C^{\text{test}}, \theta^*) p(\theta^* | r^{\text{curr}}, c^{\text{curr}}, \theta^{\text{curr}}) p(r^{\text{curr}} | c^{\text{curr}}, \theta^{\text{curr}}) \right] \tag{14}$$

We assume that $\Theta^*$ is the set of all model parameters that maximize $p(R^{\text{test}} = 1 | C^{\text{test}}, \theta^*)$.

## B    ZONE ANALYSIS

### B.1    PRELIMINARIES

For any arbitrary set $\mathcal{X}$, we use $\Delta(\mathcal{X})$ to denote the space of all probability distributions with support on $\mathcal{X}$. For any two arbitrary sets $\mathcal{X}$ and $\mathcal{Y}$, we denote the collection of all functions mapping between them as $\{\mathcal{X} \to \mathcal{Y}\} \triangleq \{f \mid f : \mathcal{X} \to \mathcal{Y}\}$.

### B.2    PROBLEM FORMULATION

A standard choice for representing a single sequential decision-making problem is the infinite-horizon, discounted Markov Decision Process (MDP) (Bellman, 1957; Puterman, 1994) defined by $\langle \mathcal{S}, \mathcal{A}, \mathcal{R}, \mathcal{T}, \mu, \gamma \rangle$. Here $\mathcal{S}$ denotes a set of states, $\mathcal{A}$ is a set of actions, $\mathcal{R} : \mathcal{S} \times \mathcal{A} \to [0, 1]$ is a deterministic reward function providing evaluative feedback signals (in the unit interval), $\mathcal{T} : \mathcal{S} \times \mathcal{A} \to \Delta(\mathcal{S})$ is a transition function prescribing distributions over next states, $\mu \in \Delta(\mathcal{S})$ is an initial state distribution, and $\gamma \in [0, 1)$ is the discount factor. Building on this formalism, our work is concerned with decision-making agents that endeavor to learn optimal behaviors across multiple tasks or goals (Kaelbling, 1993; Schaul et al., 2015), a problem we formulate as a Contextual MDP (Brunskill & Li, 2013; Hallak et al., 2015; Modi et al., 2018) (CMDP) given by $\mathcal{M} = \langle \mathcal{S}, \mathcal{A}, \mathcal{C}, \mathcal{R}, \mathcal{T}, \mu, \gamma, \chi \rangle$. Here, each task of interest is identified by an individual context contained in $\mathcal{C}$ while jointly sharing the same state-action space $\mathcal{S} \times \mathcal{A}$ and discount factor $\gamma$; meanwhile, variations in the context lead to different environment configurations that may differ in transition structure $\mathcal{T} : \mathcal{C} \times \mathcal{S} \times \mathcal{A} \to \Delta(\mathcal{S})$, reward structure $\mathcal{R} : \mathcal{C} \times \mathcal{S} \times \mathcal{A} \to [0, 1]$, and initial state distribution $\mu : \mathcal{C} \to \Delta(\mathcal{S})$.

At the beginning of an episode, a single random context is sampled $c \sim \chi(\cdot) \in \Delta(\mathcal{C})$ and held fixed as the agent contends with learning optimal behavior in the resulting MDP denoted by

$\langle \mathcal{S}, \mathcal{A}, \mathcal{R}_c, \mathcal{T}_c, \mu_c, \gamma \rangle$. At each discrete timestep $t \in \mathbb{N}$, beginning with an initial state $s_0 \sim \mu_c(\cdot)$, the agent observes the current state $s_t \in \mathcal{S}$, execute an action $a_t \in \mathcal{A}$, enjoys a reward $r_t = \mathcal{R}_c(s_t, a_t)$, and transitions to the next state $s_{t+1} \sim \mathcal{T}_c(\cdot \mid s_t, a_t)$.

Defining $\Pi \triangleq \{\mathcal{S} \times \mathcal{C} \to \Delta(\mathcal{A})\}$, a contextual policy $\pi \in \Pi$ encodes a pattern of behavior that maps the current context and state to a distribution over actions. For any fixed context $c$, the performance of an agent in the resulting MDP when starting in state $s \in \mathcal{A}$ and taking action $a \in \mathcal{A}$ is assessed by the associated action-value function $Q_c^\pi(s, a) = \mathbb{E}\left[ \sum_{t=0}^\infty \gamma^t \mathcal{R}_c(s_t, a_t) \mid s_0 = s, a_0 = a \right]$, where the expectation integrates over randomness in the action selections $a_t \sim \pi(\cdot \mid s_t, c)$ and transition dynamics $s_{t+1} \sim \mathcal{T}_c(\cdot \mid s_t, a_t)$. With the corresponding value function defined as $V_c^\pi(s) = \mathbb{E}_{a \sim \pi(\cdot \mid s_t, c)}[Q_c^\pi(s, a)]$, we slightly abuse notation and use $V_c^\pi(\mu) \triangleq \mathbb{E}_{s_0 \sim \mu(\cdot \mid c)}[V_c^\pi(s_0)]$ to integrate over the randomness in the initial state. The optimal CMDP policy $\pi^\star$ is defined as achieving supremal value $\sup_{\pi \in \Pi} \mathbb{E}_{c \sim \chi(\cdot)}[V_c^\pi(\mu)]$.

Our work operates in a general function-approximation setting where individual policies $\pi_\theta \in \Pi$ are parameterized by a vector $\theta \in \Theta \subset \mathbb{R}^d$ of arbitrary dimension $d$, for instance representing the weights of a neural network with fixed architecture. Consequently, the optimal policy $\pi_{\theta^\star}$ within this policy class $\Pi_\Theta \triangleq \{\pi_\theta \in \Pi \mid \theta \in \Theta\} \subseteq \Pi$ is defined as achieving $\sup_{\pi_\theta \in \Pi_\Theta} \mathbb{E}_{c \sim \chi(\cdot)}[V_c^{\pi_\theta}(\mu)]$. In this setting, the approximation error associated with a particular choice of policy parameterization is then given by $\mathbb{E}_{c \sim \chi(\cdot)}[V_c^\star(\mu) - V_c^{\pi_{\theta^\star}}(\mu)]$.

A priori, there is no reason to suspect that the context distribution $\chi$ an agent is charged with solving will be tailored in any sort of helpful manner to facilitate rapid or efficient learning. Intuitively, CMDPs that arise in application areas of interest will likely consist of a rich, expressive context space $\mathcal{C}$ alongside a distribution of challenging, complex tasks $\chi$ that can be easily specified by a domain expert. Consequently, the onerous burden of mastering a difficult collection of tasks with little to no scaffolding falls to the agent. This reality motivates the use of a teacher-student framework wherein the agent is viewed as a student who gradually faces tasks prescribed by a teacher. A successful teacher can incrementally synthesize a useful curriculum of tasks for the student to solve, building competency that allows to student to ultimately generalize and succeed across the original collection of challenging tasks prescribed by the distribution $\chi$.

In the next section, we provide a illustrative analysis that identifies a particular objective function for a teacher to maximize whose corresponding lower bound motivates the two key elements of the ZONE framework. Our proof techniques are inspired by the Natural Policy Gradient regret lemma of Agarwal et al. (2021) and the performance guarantee for the Policy Search by Dynamic Programming algorithm of Bagnell et al. (2003).

### B.3 A Teacher Objective for Deriving the ZONE

Recall that a function $f : \mathbb{R}^d \to \mathbb{R}$ is $\beta$-smooth if

$$||\nabla f(x) - \nabla f(x')||_2 \le \beta ||x - x'||_2 \qquad \forall x, x' \in \mathbb{R}^d.$$

A consequence of this is that $\nabla^2 f(x) \preceq \beta I, \forall x \in \mathbb{R}^d$ and so, by Taylor's Theorem,

$$|f(x') - f(x) - \nabla f(x)^\top (x' - x)| \le \frac{\beta}{2} ||x' - x||_2^2 \qquad \forall x, x' \in \mathbb{R}^d.$$

**Assumption 1.** *(Policy Smoothness) We assume that* $\log(\pi_\theta(a \mid s, c))$ *is a* $\beta$-*smooth function of* $\theta \in \Theta$, $\forall (s, a) \in \mathcal{S} \times \mathcal{A}$ *and* $c \in \mathcal{C}$.

Let $\tau = (s_0, a_0, s_1, a_1, \ldots)$ be a random trajectory sampled according to a current student policy $\pi_\theta$ under a fixed context $c \in \mathcal{C}$. Defining the advantage function as

$$A_c^\pi(s, a) = Q_c^\pi(s, a) - V_c^\pi(s),$$

we consider an abstract policy-gradient method that updates the student policy parameters based on the advantage function and a learning rate $\eta \in \mathbb{R}_{\ge 0}$ via

$$\theta' = \theta + \eta \nabla_\theta \log(\pi_\theta(a \mid s, c)) A_c^{\pi_\theta}(s, a).$$

In practice, one would choose a suitable estimator of the advantage function (Mnih et al., 2016; Schulman et al., 2016). Suppose that on-policy policy-gradient updates are performed sequentially on the state-action pairs of the sampled trajectory $\tau$ so that the student policy parameters at the beginning of the episode are $\theta^{(0)} = \theta$ and

$$\theta^{(t+1)} = \theta^{(t)} + \eta \nabla_\theta \log\left(\pi_{\theta^{(t)}}(a_t \mid s_t, c)\right) A_c^{\pi_\theta}(s_t, a_t).$$

By Assumption 1, we have that

$$\log\left(\frac{\pi_{\theta^{(t+1)}}(a_t \mid s_t, c)}{\pi_{\theta^{(t)}}(a_t \mid s_t, c)}\right) = \log\left(\pi_{\theta^{(t+1)}}(a_t \mid s_t, c)\right) - \log\left(\pi_{\theta^{(t)}}(a_t \mid s_t, c)\right)$$

$$\geq \nabla_\theta \log\left(\pi_{\theta^{(t)}}(a_t \mid s_t, c)\right)^\top \left(\theta^{(t+1)} - \theta^{(t)}\right) - \frac{\beta}{2}||\theta^{(t+1)} - \theta^{(t)}||_2^2$$

$$= \eta A_c^{\pi_\theta}(s_t, a_t)||\nabla_\theta \log\left(\pi_{\theta^{(t)}}(a_t \mid s_t, c)\right)||_2^2 - \frac{\eta^2 \beta}{2} A_c^{\pi_\theta}(s_t, a_t)^2 ||\nabla_\theta \log\left(\pi_{\theta^{(t)}}(a_t \mid s_t, c)\right)||_2^2$$

$$\geq \eta A_c^{\pi_\theta}(s_t, a_t)\left(1 - \frac{\eta\beta}{2(1-\gamma)}\right)||\nabla_\theta \log\left(\pi_{\theta^{(t)}}(a_t \mid s_t, c)\right)||_2^2,$$

where the final line leverages the fact that rewards are bounded in the unit interval implying value is upper bounded by $\frac{1}{(1-\gamma)}$. Note that this inequality only holds in this exact form for the (random) state-action pair $(s_t, a_t)$ that led to the update from policy parameters $\theta^{(t)}$ to $\theta^{(t+1)}$. Let $\rho_c^{\pi_\theta}$ denote the distribution over trajectories induced by the policy $\pi_\theta$ under context $c \in \mathcal{C}$. Let $\pi_\theta$ denote the student policy at the start of the episode and $\pi_{\theta'}$ denote the updated student policy after the episode terminates.

For brevity, we omit the state and context arguments to each policy in the following. For any trajectory $\tau = (s_0, a_0, s_1, a_1, \ldots)$, we introduce notation to denote a partial trajectory whose start and end are indexed by timesteps $i, j \in \mathbb{N}$ respectively: $\tau_i^j = (s_i, a_i, s_{i+1}, a_{i+1}, \ldots, s_{j-1}, a_{j-1}, s_j, a_j)$. With a further abuse of notation, we still use $\rho_c^\pi$ to denote the distribution over such partial trajectories sampled while executing policy $\pi$ in the MDP induced under context $c \in \mathcal{C}$. An objective for the teacher $\Lambda \in \Delta(\mathcal{C})$ is

$$\max_{\Lambda \in \Delta(\mathcal{C})} \mathbb{E}_{c \sim \Lambda(\cdot)}\left[\sum_{t=0}^\infty \mathbb{E}_{\tau_0^{t-1} \sim \rho_c^{\pi_{\theta^\star}}(\cdot)}\left[\mathbb{E}_{s_t \sim \mathcal{T}_c(\cdot | s_{t-1}, a_{t-1})}\left[D_{\mathrm{KL}}(\pi_{\theta^\star} \mid\mid \pi_{\theta^{(t)}}) - D_{\mathrm{KL}}(\pi_{\theta^\star} \mid\mid \pi_{\theta^{(t+1)}})\right]\right]\right].$$

At a high level, this says that a good teacher prescribes a distribution over environments for a fixed student such that the resulting policy updates bring the student closer to the optimal policy in $\Pi_\theta$. In slightly more detail, this is achieved by examining rollouts of increasing lengths generated by the optimal policy $\pi_{\theta^\star}$ and assessing the reduction in KL-divergence between the student policy and the optimal policy before and after the policy-gradient update. For brevity, we continue onward assuming a fixed context $c \in \mathcal{C}$, allowing us to drop the outermost expectation.

Let $\mathcal{X}$ be an arbitrary set consider any two distributions $\nu, \nu' \in \Delta(\mathcal{X})$. Recall that the total variation distance is an integral probability metric (Müller, 1997; Sriperumbudur et al., 2009) defined as

$$D_{\mathrm{TV}}(\nu \mid\mid \nu') = \sup_{f \in \mathcal{F}} |\mathbb{E}_{x \sim \nu(\cdot)}[f(x)] - \mathbb{E}_{x \sim \nu'(\cdot)}[f(x)]|, \qquad \mathcal{F} = \{f : \mathcal{X} \to \mathbb{R} \mid ||f||_\infty \leq 1\}.$$

Consequently, for any function $f : \mathcal{X} \to \mathbb{R}$ such that $||f||_\infty \leq C < \infty$, it follows that

$$|\mathbb{E}_{x \sim \nu(\cdot)}[f(x)] - \mathbb{E}_{x \sim \nu'(\cdot)}[f(x)]| \leq C \cdot D_{\mathrm{TV}}(\nu \mid\mid \nu').$$

In order to apply this fact to induce a distribution shift, we make the following assumption which controls for the variability in log-likelihood ratio between two policies separated by a single policy-gradient update:

**Assumption 2.** *(Bounded log-likelihood ratio) Let $\theta$ be an initial set of policy parameters and $\theta'$ denote the policy parameters after a single policy-gradient update. For any fixed $c \in \mathcal{C}$, we assume that there exists a numerical constant $C < \infty$ such that $\log\left(\frac{\pi_{\theta'}(a|s,c)}{\pi_\theta(a|s,c)}\right) \leq C, \forall (s, a) \in \mathcal{S} \times \mathcal{A}.$*

Expanding from above, we have

$$\sum_{t=0}^{\infty} \mathbb{E}_{\tau_0^{t-1} \sim \rho_c^{\pi_{\theta^\star}}(\cdot)} \left[ \mathbb{E}_{s_t \sim \mathcal{T}_c(\cdot | s_{t-1}, a_{t-1})} \left[ D_{\mathrm{KL}}(\pi_{\theta^\star} \| \pi_{\theta(t)}) - D_{\mathrm{KL}}(\pi_{\theta^\star} \| \pi_{\theta(t+1)}) \right] \right]$$

$$= \sum_{t=0}^{\infty} \mathbb{E}_{\tau_0^{t-1} \sim \rho_c^{\pi_{\theta^\star}}(\cdot)} \left[ \mathbb{E}_{s_t \sim \mathcal{T}_c(\cdot | s_{t-1}, a_{t-1})} \left[ \mathbb{E}_{a_t \sim \pi_{\theta^\star}(\cdot | s_t, c)} \left[ \log \left( \frac{\pi_{\theta(t+1)}(a_t | s_t, c)}{\pi_{\theta(t)}(a_t | s_t, c)} \right) \right] \right] \right]$$

$$= \sum_{t=0}^{\infty} \mathbb{E}_{\tau_0^{t} \sim \rho_c^{\pi_{\theta^\star}}(\cdot)} \left[ \log \left( \frac{\pi_{\theta(t+1)}(a_t | s_t, c)}{\pi_{\theta(t)}(a_t | s_t, c)} \right) \right]$$

$$= \sum_{t=0}^{\infty} \mathbb{E}_{\tau \sim \rho_c^{\pi_{\theta^\star}}(\cdot)} \left[ \log \left( \frac{\pi_{\theta(t+1)}(a_t | s_t, c)}{\pi_{\theta(t)}(a_t | s_t, c)} \right) \right]$$

$$= \mathbb{E}_{\tau \sim \rho_c^{\pi_{\theta^\star}}(\cdot)} \left[ \sum_{t=0}^{\infty} \log \left( \frac{\pi_{\theta(t+1)}(a_t | s_t, c)}{\pi_{\theta(t)}(a_t | s_t, c)} \right) \right]$$

$$\geq \mathbb{E}_{\tau \sim \rho_c^{\pi_{\theta^\star}}(\cdot)} \left[ \sum_{t=0}^{\infty} \gamma^t \log \left( \frac{\pi_{\theta(t+1)}(a_t | s_t, c)}{\pi_{\theta(t)}(a_t | s_t, c)} \right) \right]$$

$$\geq \mathbb{E}_{\tau \sim \rho_c^{\pi_{\theta}}(\cdot)} \left[ \sum_{t=0}^{\infty} \gamma^t \log \left( \frac{\pi_{\theta(t+1)}(a_t | s_t, c)}{\pi_{\theta(t)}(a_t | s_t, c)} \right) \right] - \frac{C}{(1-\gamma)} \cdot D_{\mathrm{TV}}(\rho_c^{\pi_{\theta^\star}} \| \rho_c^{\pi_\theta})$$

$$\geq \eta \left( 1 - \frac{\eta\beta}{2(1-\gamma)} \right) \mathbb{E}_{\tau \sim \rho_c^{\pi_\theta}(\cdot)} \left[ \sum_{t=0}^{\infty} \gamma^t A_c^{\pi_\theta}(s_t, a_t) \|\nabla_\theta \log \left( \pi_{\theta(t)}(a_t | s_t, c) \right) \|_2^2 \right] - \frac{C}{(1-\gamma)} \cdot D_{\mathrm{TV}}(\rho_c^{\pi_{\theta^\star}} \| \rho_c^{\pi_\theta})$$

$$\geq \eta \left( 1 - \frac{\eta\beta}{2(1-\gamma)} \right) \mathbb{E}_{\tau \sim \rho_c^{\pi_\theta}(\cdot)} \left[ \sum_{t=0}^{\infty} \gamma^t A_c^{\pi_\theta}(s_t, a_t) \|\nabla_\theta \log \left( \pi_{\theta(t)}(a_t | s_t, c) \right) \|_2^2 \right] - \frac{C}{\sqrt{2}(1-\gamma)} \cdot \sqrt{D_{\mathrm{KL}}(\rho_c^{\pi_{\theta^\star}} \| \rho_c^{\pi_\theta})},$$

where the first inequality follows as $\gamma \in [0, 1]$, the second inequality leverages Assumption 2 and the aforementioned fact to shift trajectory distributions, the penultimate inequality applies our earlier policy-gradient norm lower bound, and the final inequality follows as Pinsker's inequality (Pinsker, 1964).

Now accounting for the randomness in the contexts, we have the following lower bound to the teacher (maximization) objective:

$$\mathbb{E}_{c \sim \Lambda(\cdot)} \left[ \sum_{t=0}^{\infty} \mathbb{E}_{\tau_0^{t-1} \sim \rho_c^{\pi_{\theta^\star}}(\cdot)} \left[ \mathbb{E}_{s_t \sim \mathcal{T}_c(\cdot | s_{t-1}, a_{t-1})} \left[ D_{\mathrm{KL}}(\pi_{\theta^\star} \| \pi_{\theta(t)}) - D_{\mathrm{KL}}(\pi_{\theta^\star} \| \pi_{\theta(t+1)}) \right] \right] \right]$$

$$\geq \mathbb{E}_{c \sim \Lambda(\cdot)} \left[ \underbrace{\eta \left( 1 - \frac{\eta\beta}{2(1-\gamma)} \right) \mathbb{E}_{\tau \sim \rho_c^{\pi_\theta}} \left[ \sum_{t=0}^{\infty} \gamma^t A_c^{\pi_\theta}(s_t, a_t) \|\nabla_\theta \log \left( \pi_{\theta(t)}(a_t | s_t, c) \right) \|_2^2 \right]}_{\text{①}} - \underbrace{\frac{C}{\sqrt{2}(1-\gamma)} \cdot \sqrt{D_{\mathrm{KL}}(\rho_c^{\pi_{\theta^\star}} \| \rho_c^{\pi_\theta})}}_{\text{②}} \right]$$

Observe that term ① captures a notion of learning potential for the current student under tasks sampled by the teacher. Intuitively, this is quantified by looking at the average, advantage-weighted policy-gradient norm across trajectories generated by the student policy $\pi_\theta$ in each sampled context $c \sim \Lambda(\cdot)$. Orthogonally, term ② encapsulates a notion of problem difficulty as measured by how much the trajectory distribution of the student differs from that of the optimal policy in each sampled context.

When this latter quantity ② is too large, suggesting an overwhelmingly difficult problem for the student where a large number of samples or environment interactions will be needed to improve performance, this term overpowers any learning potential captured in term ①. Conversely, tasks that are too easy for the student will lead to scenarios where both ② and the advantage terms encountered in ① will be small (or even zero), suggesting little opportunity for improving performance. Naturally, the "sweet spot" or ZPD suggested by this lower bound consists of a teacher selecting tasks with

reasonably large policy gradient norms (signaling learning potential) while being within the students means (as measured by the divergence between the student's trajectory distribution from that of the optimal policy). Practical approaches to automated curriculum design use a notion of pseudo-regret in lieu of ②, accounting for a lack of knowledge about the optimal policy in advance (Florensa et al., 2018b; Dennis et al., 2020).

# C   ADDITIONAL TEACHER ANALYSIS

Here we include additional analysis on the teacher.

## C.1   PAIRED ON MUJOCO ENVIRONMENTS

Figure 6 shows the rejection rate and student's gradient norms. Interestingly, REJECT tends to have higher rejection rates. In general, GRAD and PAIRED have similar rejection rates.

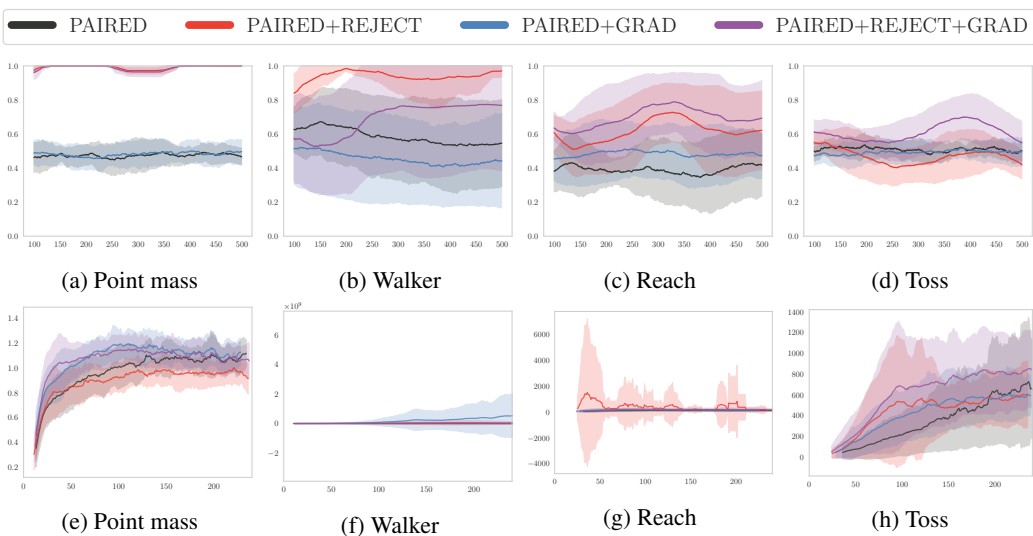

Figure 6: On the MuJoCo environments. The rejection rate is reported in the first row (a-d), and the gradient norms are reports in the second row (e-h).

## C.2   GOAL GAN

Figure 7 shows the rejection rates on the Goal GAN environments and the MuJoCo environments. In general, the rejection rates are similar across environments.

# D   ALGORITHM INFO

We provide information on the PAIRED and Goal GAN implementations.

## D.1   PAIRED ON MINIGRID ENVIRONMENTS

We use the implementation at `https://github.com/ucl-dark/paired` which is based on Dennis et al. (2020)'s implementation. We do not change any hyperparameters in their algorithm. All hyperparameters are the same as those reported in Dennis et al. (2020). We refer to their paper for more details on PAIRED. We run all the variants of PAIRED with `ZONE` with 10 seeds.

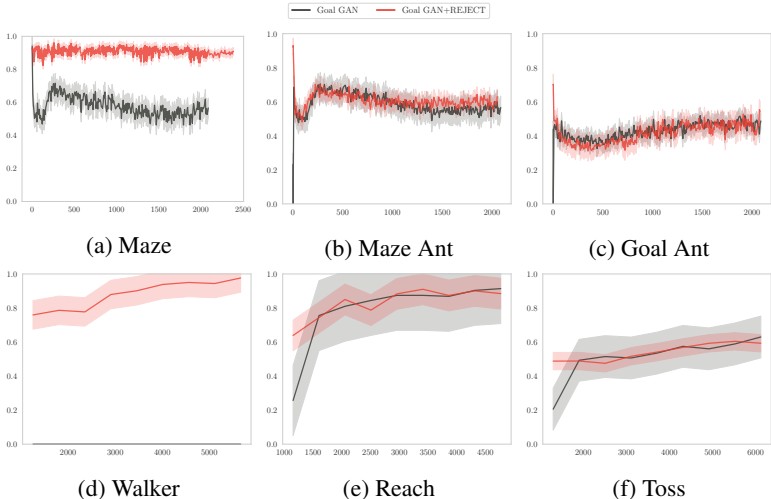

Figure 7: Reported is the rejection rate. The rate for Goal GAN on the original Goal GAN environments is reported in the first row, and the rate on the MuJoCo environments is reported in the last row.

## D.2 GOAL GAN ON GOAL GAN ENVIRONMENTS

We use the original implementation at `https://github.com/florensacc/rllab-curriculum`. We do not change any hyperparameters. We refer to their paper (Florensa et al., 2018b) for more details on Goal GAN. We run all the variants of Goal GAN with `ZONE` with 5 seeds.

## D.3 PAIRED AND GOAL GAN ON MUJOCO ENVIRONMENTS

We use Du et al. (2022)'s implementation of PAIRED and Goal GAN on their MuJoCo environments. Their MuJoCo setup is designed to test students on out-of-domain goals, that the students have not yet seen. We run all the variants of both algorithms with `ZONE` with 10 seeds. We do not change any of the hyperparameters. We use the default Goal GAN implementation in their work. Goal GAN stores 500 goals and the student is evaluated on the goals 3 times. The mean reward is taken and used to determine the label for the teacher. The label is 1 if the mean reward lies within the difficulty criterion ($r \in [0.1, 0.9]$), and 0 otherwise. Every 500 steps, the teacher trains on the labelled data.

PAIRED is implemented using the default parameters from Du et al. (2022)'s algorithms but with symmetrization turned off (ie. we remove the second teacher in their work).

## E DO MEASURES OF DIFFICULTY MATTER?

A key component to `ZONE` is the choice of difficulty measure. Most prior work use reward to model difficulty: The lower of the reward, the more difficulty the problem. If the problem is more difficult, then the student is less likely to succeed on the problem.

However, we find that using *dense* rewards as a proxy measure for difficulty is misleading. Dense rewards are typically used for training students in the MuJoCo control setting, as done in Du et al. (2022). Running `ZONE` on PAIRED in this setting reveals an interesting discrepancy: `ZONE` can achieve higher episodic reward than the base algorithm (Figure 8a), however achieves low success (Figure 8b). For example, at 2000 steps, episode reward scores increasingly higher from PAIRED, PAIRED+GRAD, PAIRED+REJECT , to PAIRED+REJECT +GRAD. However, success scores increasingly higher from PAIRED+GRAD, PAIRED+REJECT , PAIRED+REJECT +GRADto PAIRED.

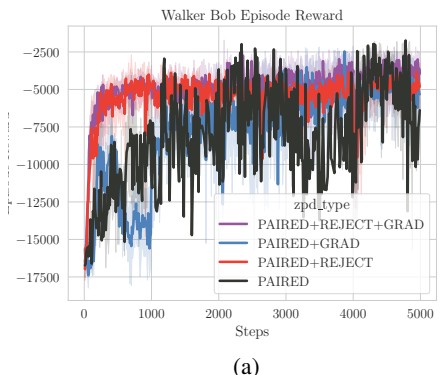 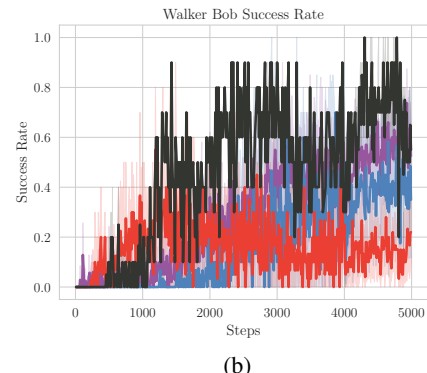

(a)                                        (b)

Figure 8: PAIRED on the Walker domain. The figures show the performance of the student on the out-of-domain goals. (a) shows the student's test return over the course of training. (b) shows the student's success rate over the course of training.

This discrepancy reveals that training the teacher based on a dense-reward difficulty measure can be misleading when the reward function does not correlate well with the student's success. ZONE is sensitive to this choice of difficulty measure that is not well correlated with success. Thus, for MuJoCo experiments, we choose to use the student's success as a measure of difficulty which is what PAIRED in Dennis et al. (2020) and Goal GAN assume. This gives us the results from the previous section in Figure 3 and Figure 4.

# F   ADDITIONAL IMAGES OF GENERATED PAIRED ENVIRONMENTS

See Figure 2-7.

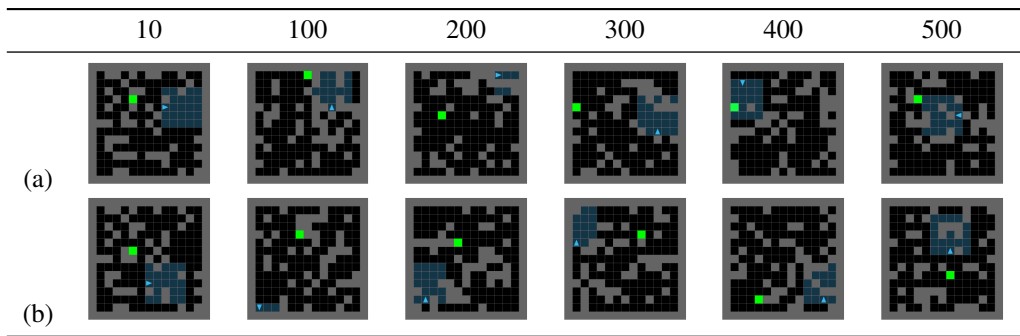

Table 2: (a) row are tasks generated from PAIRED, and (b) row are tasks generated from PAIRED+REJECT+GRAD. The columns mark the number of model updates. This table includes updates 10-500.

# G   ABLATION ON GRAD

See Figure 9.

| | 600 | 700 | 800 | 900 | 1000 | 1100 |
|---|---|---|---|---|---|---|

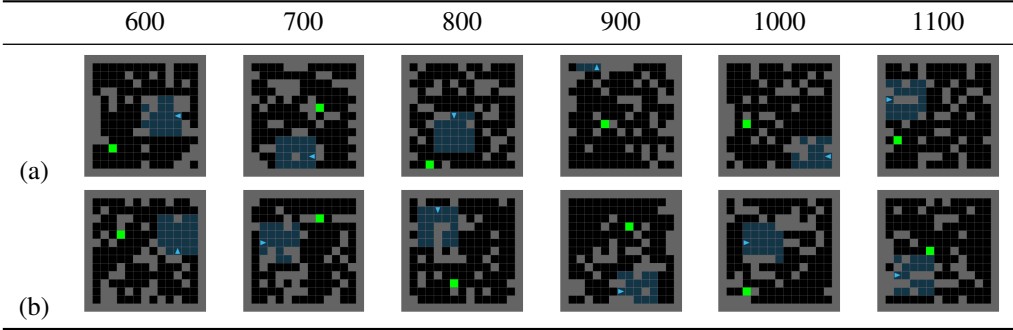

Table 3: (a) row are tasks generated from PAIRED, and (b) row are tasks generated from PAIRED+REJECT+GRAD. The columns mark the number of model updates. This table includes updates 600-1100.

| | 1200 | 1300 | 1400 | 1500 | 1600 | 1700 |
|---|---|---|---|---|---|---|

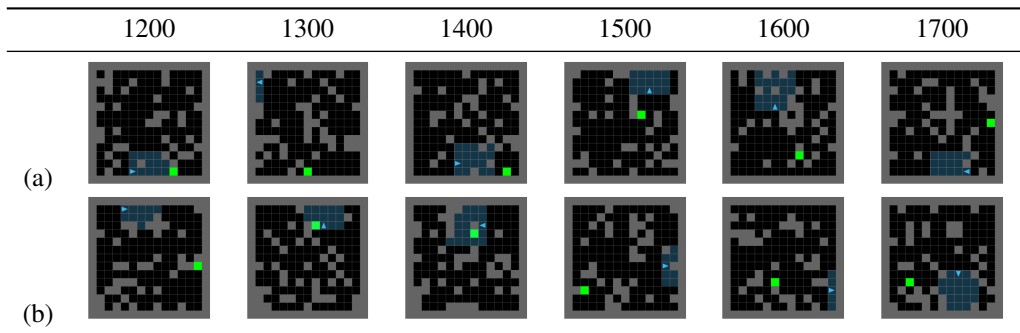

Table 4: (a) row are tasks generated from PAIRED, and (b) row are tasks generated from PAIRED+REJECT+GRAD. The columns mark the number of model updates. This table includes updates 1200-1700.

| | 1800 | 1900 | 2000 | 2100 | 2200 | 2300 |
|---|---|---|---|---|---|---|

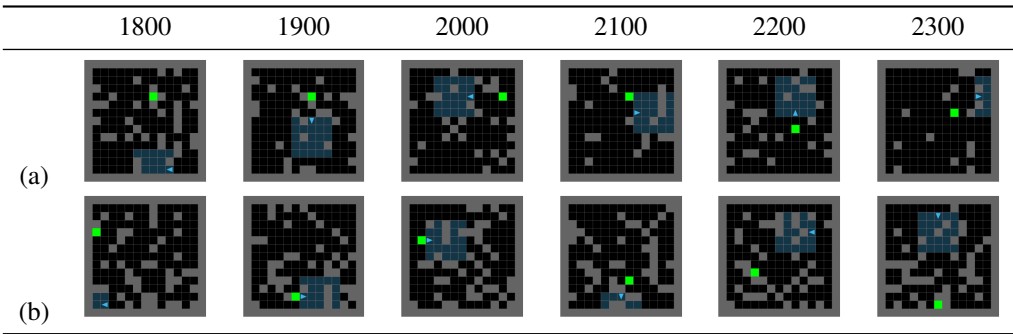

Table 5: (a) row are tasks generated from PAIRED, and (b) row are tasks generated from PAIRED+REJECT+GRAD. The columns mark the number of model updates. This table includes updates 1800-2300.

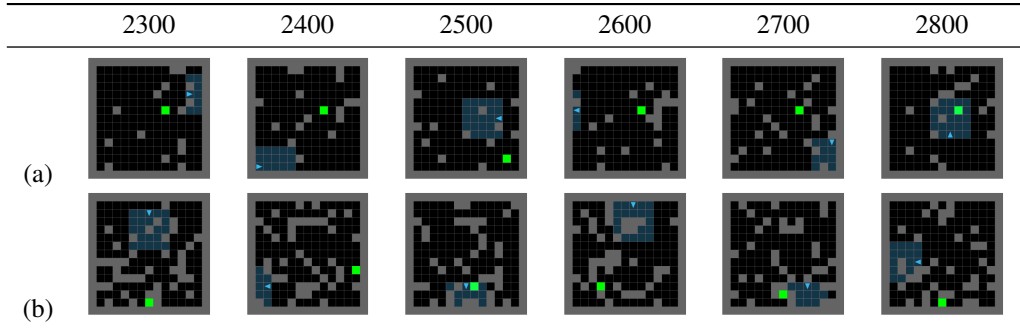

Table 6: (a) row are tasks generated from PAIRED, and (b) row are tasks generated from PAIRED+REJECT+GRAD. The columns mark the number of model updates. This table includes updates 2300-2800.

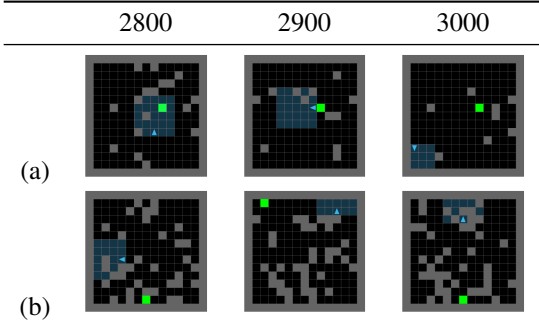

Table 7: (a) row are tasks generated from PAIRED, and (b) row are tasks generated from PAIRED+REJECT+GRAD. The columns mark the number of model updates. This table includes updates 2800-3000.

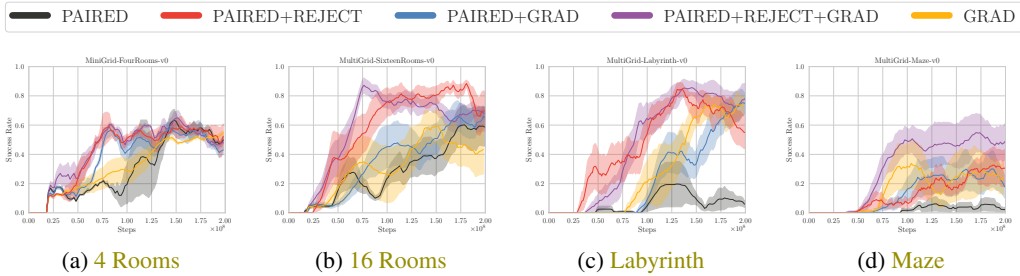

(a) 4 Rooms          (b) 16 Rooms          (c) Labyrinth          (d) Maze

Figure 9: PAIRED performance on MiniGrid transfer environments which includes an ablation of just training the PAIRED teacher with GRAD (ie. just training the teacher with the student's gradient norms). Note that GRAD is run on 4 seeds, and the other methods are run on 10 seeds.

