# OpenReview forum: "In the ZONE: Measuring difficulty and progression in curriculum generation"
_ICLR.cc/2023/Conference — Submitted to ICLR 2023_

### Official Review · Reviewer_BAdD · 2022-10-20

**Confidence:** 4
**Clarity, Quality, Novelty And Reproducibility:** Yes
**Correctness:** 3
**Technical Novelty And Significance:** 3
**Empirical Novelty And Significance:** 2
**Recommendation:** 5

**Details Of Ethics Concerns:**

No concern.

**Strength And Weaknesses:**

Strengths:

1. The proposed ideas are simple and intuitive.
2. The idea of using the amount of student model updates to guide the teacher is interesting. While there are some similar ideas in curriculum learning, it is nice to see it is effective in RL.

Weaknesses:

1. It is unclear to me why REJECT helps. It simply drops the tasks that are too difficult or easy. However, it still requires generating data with the student model. So it is still expensive. I am also curious whether such dropped samples are counted in the learning curve.
2. The improvement over the baselines is not significant. It is only comparable with goal GAN.
3. While the focus of the paper is curriculum learning, the paper should consider baselines that use intrinsic rewards in the experiment. The selected environments are mainly hard-exploration environments, so intrinsic reward methods (such as count-based intrinsic reward) could also perform well in these environments. So I would like to see a comparison.
4. Various exploration methods (such as intrinsic rewards) are not discussed in the related work. They try to tackle similar challenges in different ways.

**Summary Of The Paper:**

This paper studies curriculum learning in reinforcement learning. To achieve this, the paper formulates a teacher-student framework, where the teacher is responsible to generate the tasks for the student to learn. The key idea is two-folded. First, the student should not learn from the tasks that are outside of the difficulty scope. Second, the teacher should be rewarded for generating tasks that cause larger parameters update of the policy/value networks of the student. To implement this idea, the paper proposes REJECT, which simply rejects the tasks that are too easy or too hard based on the rollout reward, and GRAD, which makes the amount of the parameters update as one of the rewards for the teacher. Experiments show that the two strategies can improve efficiency.

**Summary Of The Review:**

This paper presents some interesting ideas. However, I have concerns about the REJECT. Particularly, it is unclear to me whether the dropped samples are counted when drawing the learning curve. I would like to ask the authors to clarify. If not counted, the comparison could be unfair. The experiments also need improvement.

---

> ### Author Response · Authors · 2022-11-11
> **Thank you for your feedback! We have revised the paper to address the reviewer's concerns.**
>
> We thank the reviewer for their feedback and suggestions. We are glad that the reviewer finds our method intuitive and the idea of using the student’s gradient norms for RL curriculum learning interesting. We’ve revised the paper to improve clarity on the setup, REJECT’s improvements to the teacher’s and student’s performance, and the connection to curriculum learning algorithms for generating intrinsic rewards. Our revisions are marked in orange.
>
> > it is unclear to me whether the dropped samples are counted when drawing the learning curve.
>
> Thanks for asking this! **The dropped samples are counted in the learning curve. We’ve revised the caption to this Figure to emphasize that the curves see the same amount of data at a given x-value.** As we note on page 5 under “Modeling … with rejection sampling”, we upsample the accepted tasks to replace the removed tasks. Each curve on the graph sees the same amount of training data at a given x value. Therefore, the comparisons are fair.
>
> > It is unclear to me why REJECT helps. It simply drops the tasks that are too difficult or easy.
>
> REJECT helps both the teacher generate better tasks and the student train directly on the most relevant tasks. Below, we elaborate this in the context of PAIRED’s regret formalism and the current evidence presented in the paper (visualizations in Table 1, the student’s learning performance inFigure 3-4 and formal analysis in Appendix B).
>
> In PAIRED, **REJECT helps the student improve because the teacher targets tasks where the student still needs to learn an optimal sequence of actions. REJECT helps the teacher improve because the teacher directly trains on tasks that incur positive regret.** Tasks that are rejected are ones with negative/zero regret: These are tasks where the student does better than (or equal to) the anti-student. In sparse reward settings, these tasks do not present large learning opportunities for the student. Therefore, if the teacher only keeps tasks with positive regret, it curates tasks for the student where the student has a larger learning opportunity. Directly training the teacher on the accepted problems results in the teacher learning to generate better tasks much faster (rf. Figure 5b, rejection rates).
>
> **Our existing analysis in the paper illustrates these points as well.** REJECT aids the teacher to generate tasks that do not inhibit student learning (tasks that are too difficult) and are not yet easily mastered by the student (tasks that are too difficult). Our paper illustrates this point in Table 1 where we compare the environments that are generated from PAIRED and the environments that are kept after rejection sampling. Noticeably, the environments from PAIRED are much easier than the ones from our method, leading to the fast generalization performance seen in the previous Figure 3.
>
> **Additionally, we provide a formal analysis in Appendix B.** Our formal analysis in Appendix B suggests that REJECT measures how much the trajectory distribution of the student differs from that of the optimal policy in each sampled context. Training on problems that only fall inside the student’s difficulty range avoids tasks where a large number of samples or environment interactions will be needed to improve performance, but also where there’s little opportunity for improving performance.
>
> >  [REJECT] still requires generating data with the student model. So it is still expensive.
>
> **REJECT does not require additional computation compared to normal PAIRED or GoalGAN.** As we note on page 5 under “Modeling … with rejection sampling,” we are not running the teacher or student more often when we do rejection sampling: We upsample the accepted tasks to replace the removed tasks. Therefore, REJECT requires the same amount of computation time as having no REJECT. We have revised the paper to make this important point of avoiding additional computational cost more clear.
>
> > The improvement over the baselines is not significant. It is only comparable with goal GAN.
>
> **The improvement over baselines for PAIRED is significant (Figure 3).** We agree with the reviewer that REJECT is less helpful on Goal GAN’s control environments; we note this already in the experiments section, and additionally elaborate: Goal GAN environments are much simpler navigation environments compared to the Minigrid environments that do not need much of a curriculum. **We chose to additionally include settings where the ZONE techniques do comparably well or slightly better than prior work because these settings illustrate where these techniques help. **
>
> **ZONE does lead to large improvements in settings where the task structure and task generations are complex** (eg. Minigrid environments). In summary, ZONE leads to significant improvements on complex tasks, and not so much on continuous control environments where agents are already trained with dense rewards (eg. the MuJoCo environments) or where the navigation task is already easy.

---

> > ### Author Response · Authors · 2022-11-11
> > **Continuation of revisions**
> >
> > > While the focus of the paper is curriculum learning, the paper should consider baselines that use intrinsic rewards in the experiment.
> >
> > **Our method is not an exploration method. Our paper focuses on the challenge of generalizing on out-of-distribution domains, and emphasizes the effects of good teaching on this challenge via a curriculum.** Prior intrinsic reward works focus on testing on in-distribution environments [eg. 1,2], because the focus of those works is on how to better explore a given environment. Our work focuses on curriculum generation algorithms where a student only learns from a teacher, and the teacher is focused on generating tasks that can help the student generalize more effectively to novel tasks. The baselines and our method can also be augmented with intrinsic rewards, however this is not within the paper’s curriculum learning scope.
> >
> > > Various exploration methods (such as intrinsic rewards) are not discussed in the related work. They try to tackle similar challenges in different ways.
> >
> > Thanks for the suggestion! Our current related work section does already include curriculum learning works applied to intrinsic rewards (eg. [3,4]). As mentioned in our previous response, prior intrinsic reward works do not look at the challenge of generalizing on out-of-distribution environments—they tackle a different challenge of exploration! We believe an interesting future direction might be to unify the curriculum learning works that focus on the generalization challenge and on the exploration challenge.
> >
> > [1] Deepak Pathak, Pulkit Agrawal, Alexei A. Efros and Trevor Darrell. Curiosity-driven Exploration by Self-supervised Prediction. In ICML 2017.
> > [2] Ostrovski, Georg, et al. "Count-based exploration with neural density models." International conference on machine learning. PMLR, 2017.
> > [3] Campero, Andres, et al. "Learning with AMIGo: Adversarially Motivated Intrinsic Goals." International Conference on Learning Representations. 2020.
> > [4] Sukhbaatar, Sainbayar, et al. "Intrinsic Motivation and Automatic Curricula via Asymmetric Self-Play." International Conference on Learning Representations. 2018.

---

> > > ### Comment · Reviewer_BAdD · 2022-11-11
> > > **Thank you for the clarification and response**
> > >
> > > My biggest concern on the fairness of comparison is resolved by the clarification. I will increase the score. However, I still intend to not accept the paper because 1) I am not convinced that the improvement is significant, and 2) the paper lacks a discussion of exploration methods for out-of-distribution generalization.
> > >
> > > Since the improvement is claimed to be mainly in the Minigrid environments, the proposed method should be evaluated against the state-of-the-art baselines in the Minigrid domain, e.g., [1] [2] [3] [4]. In particular, [1] also uses a teacher-student framework so it should be compared.
> > >
> > > ```
> > > prior intrinsic reward works do not look at the challenge of generalizing on out-of-distribution environments
> > > ```
> > > This is inaccurate. For example, [2] [3] are tailored for the exploration in generalizing on out-of-distribution environments. While I understand that, the teacher-student framework differs from intrinsic rewards methods, their outcome on the experimented environments is similar, i.e., encouraging exploration. Thus, the paper should provide a more comprehensive discussion of the exploration methods, especially for the ones that can generalize on out-of-distribution environments.
> > >
> > >
> > >
> > >
> > >
> > >
> > >
> > >
> > > [1] Campero, Andres, et al. "Learning with amigo: Adversarially motivated intrinsic goals." arXiv preprint arXiv:2006.12122 (2020).
> > >
> > > [2] Raileanu, Roberta, and Tim Rocktäschel. "RIDE: Rewarding Impact-Driven Exploration for Procedurally-Generated Environments." (2020).
> > >
> > > [3] Zha, Daochen, et al. "Rank the Episodes: A Simple Approach for Exploration in Procedurally-Generated Environments." International Conference on Learning Representations. 2020.
> > >
> > > [4] Zhang, Tianjun, et al. "Noveld: A simple yet effective exploration criterion." Advances in Neural Information Processing Systems 34 (2021): 25217-25230.

---

> > > > ### Author Response · Authors · 2022-11-15
> > > > **Thanks for the comments! We've added further discussion on the intrinsic reward literature.**
> > > >
> > > > Thanks for your comments and suggestions! We’ve added a further discussion in the paper (see Section 2 under teacher-student curriculum generation) which adds the papers the reviewer has suggested and discusses existing exploration methods that test on OOD environments. Please let us know if you have further recommendations!
> > > >
> > > > > 1) I am not convinced that the improvement is significant
> > > >
> > > > As mentioned previously, we would like to highlight that our method provides significant improvements above PAIRED. We agree with the reviewer and acknowledge that the improvements above GoalGAN are not significant. We want to be honest in providing evidence and hypotheses for when ZONE does lead to significant improvements in curriculum generation algorithms. This is why we include sets of experiments across discrete and continuous environments, all used in prior curriculum generation works. We hope this will not overly influence your impression of the remaining significant results!

---

> > > > > ### Comment · Reviewer_BAdD · 2022-11-21
> > > > > **Thank you for the further reponse**
> > > > >
> > > > > Thank you for the response. But I am still not convinced that the improvement is significant. It will be more convincing if using the state-of-the-art (intrinsic rewards or curriculum learning) base agent in Minigrid to show that ZONE can also achieve improvement.

---

### Official Review · Reviewer_9WXc · 2022-10-23

**Confidence:** 2
**Correctness:** 3
**Technical Novelty And Significance:** 2
**Empirical Novelty And Significance:** 2
**Recommendation:** 5

**Clarity, Quality, Novelty And Reproducibility:**

The paper is clear, but I am not confident about the novelty of the proposed method.

**Strength And Weaknesses:**

Strengths:
- The paper is well written.

Weaknesses:
- Experimental results are mixed (sometimes better, sometimes worse). Thus, the benefit of the proposed method is not clearly shown.
- The title should be clearly tied to reinforcement learning. The title makes it look like the idea is more generic, while being tested only on RL. If the authors prefer to keep the title as is, they should demonstrate applicability in other areas, e.g. classification.
- There are many competing curriculum learning approaches for RL, see [A], missed by the authors in their discussion.

[A] Soviany et al., "Curriculum learning: A survey", International Journal of Computer Vision, 2022.


**Summary Of The Paper:**

The authors study to problem of finding and using the right balance in curriculum learning applied to reinforcement learning (RL). More specifically, at each training iteration of a student network, the proposed method aims to generate an appropriate, i.e. not too easy, not too hard, task for the student to learn. The goal is to achieve faster convergence and better performance. The approach is applied on top of two existing methods and tested on two benchmarks.

**Summary Of The Review:**

Due to the mixed results and the missed related works, I am not confident about accepting the paper.

---

> ### Author Response · Authors · 2022-11-11
> **Thank you for your feedback! We have revised the paper to address the reviewer's concerns.**
>
> We thank the reviewer for their feedback and suggestions. We have updated the paper to address the reviewer’s concerns.  Our revisions are marked in orange.
>
> > Experimental results are mixed (sometimes better, sometimes worse). Thus, the benefit of the proposed method is not clearly shown.
>
> We’d like to emphasize that we already note significant gains on the hard environments (Minigrid) and comparable/slightly better gains on the easier environments (control environments). As elaborated in Section 5.1, the PAIRED teacher benefits much more from ZONE because it has a much harder learning problem than the teacher in the control environments: the PAIRED teacher has to generate an entire $15 \times 15$ grid with up to $50$ walls, whereas the GoalGAN teacher generates only a 2D goal. It is much harder for the PAIRED teacher to control for difficulty as small variations in its generations can result in large variations of the task difficulty. For example, placing a single wall piece at the end of a corridor might block off a direct route to the goal for the student. By contrast, in the Goal GAN/control environments which typically have little to no obstacles, small variations the 2D goals would not make a large difference in the task difficulty.
>
> > The title should be clearly tied to reinforcement learning. The title makes it look like the idea is more generic, while being tested only on RL. If the authors prefer to keep the title as is, they should demonstrate applicability in other areas, e.g. classification.
>
> Thanks for the suggestion! We are hesitant to revise the title to include “reinforcement learning” as this would make the title quite long. Additionally, we’d like to note that several other RL curriculum learning works do not tie their titles to RL either [1,2,3,4].
>
> [1] Yuqing Du, Pieter Abbeel, and Aditya Grover. It takes four to tango: Multiagent self play for automatic curriculum generation. In International Conference on Learning Representations, 2022
>
> [2] Dennis, Michael, et al. "Emergent complexity and zero-shot transfer via unsupervised environment design." Advances in neural information processing systems 33 (2020): 13049-13061.
>
> [3] Matiisen, Tambet, et al. "Teacher–student curriculum learning." IEEE transactions on neural networks and learning systems 31.9 (2019): 3732-3740.
>
> [4] Sukhbaatar, Sainbayar, et al. "Intrinsic motivation and automatic curricula via asymmetric self-play." arXiv preprint arXiv:1703.05407 (2017).
>
> > There are many competing curriculum learning approaches for RL, see [A], missed by the authors in their discussion.
>
> Thanks for the suggestion. Our related works section already covers several of the approaches mentioned in [A], but we’re happy to additionally include [A] in our citations!

---

### Official Review · Reviewer_3apF · 2022-10-24

**Confidence:** 4
**Correctness:** 3
**Technical Novelty And Significance:** 2
**Empirical Novelty And Significance:** 2
**Recommendation:** 5

**Clarity, Quality, Novelty And Reproducibility:**

The paper lacks enough novelty and needs more clarification on the motivation. Meanwhile, more clarifications on the setting, modelling and experiments are needed.

**Strength And Weaknesses:**

The paper develops an analytical objective for task generation in Equation (1)-(5), clearly defining the problem. Also, the two techniques are well described, and the proposed algorithm is easy to follow and easy to apply. The experiments are well designed, including comparisons with two representative baselines and analysis of generated tasks.

However, there are still some improvements that could be made to the paper:

(1) This paper does not bring in enough novelty. The problem is controlling the difficulty of tasks in the zone of proximal development is already considered in the two baselines. There is no new approach to control the difficulty proposed. The first proposed technique, REJECT, is based on the developed methodologies of the baselines and is already developed in the algorithm POET, proposed by Wang et al. (2019). POET already proposes to reject or delete new environments that are too hard or too easy for the agent.

(2) The motivation of the approach needs more clarification. Neither of the two proposed techniques is proven to maximize the derived objective. Meanwhile, the paper is inspired by generating tasks in the zone of proximal development, which is not too hard nor too easy for the student. However, maximizing the student's gradient norm is unrelated to this main idea of generating ZPD tasks.

(3) More description of the setting is needed. It is unclear whether the paper focuses on a generalization objective to maximize expected returns among a given environment distribution or what.

(4) Modelling the probability in the objective in section 3.3 on page 5 is unclear and technically incorrect. The paper claims that the first technique, REJECT, models the probability $p(r_{curr}|c_{curr},\theta_{curr})$ but how? Furthermore, this probability is decided by the dynamics of the MDP and cannot be influenced by the teacher. This modelling claim is technically incorrect.

(5) The test on the gradient maximization idea is limited. Some ablation studies to test the gradient maximization idea alone would be interesting, for example, whether gradient maximization is better than regret maximization proposed by PAIRED and difficulty control by Goal GAN.

(6) Some descriptions of experiments are missing. For example, how to set the gradient norm and returns scale when applying the gradient maximization technique? what contexts are all algorithms tested on for tasks (a)-(g) in Figure 2 on page 7? How the rejection rate is computed in Figure 5 on page 9?

(7) I have a few questions about the experimental results:

(a) We see improvements in PAIRED-based algorithms. But PAIRED generates tasks which maximize the current regret. So why are there so many tasks given by the PAIRED rejected? Does PAIRED generate many negative regret tasks?

(b) It is shown in Table 1 on page 9 that PAIRED+REJECT+GRAD gives hard tasks from the beginning. Does it break the increasing difficulty set by the curriculum learning? Is it good?

(c) REJECT does not improve Goal GAN much. Can we say PAIRED is not controlling the difficulty of tasks well as claimed, but Goal GAN already fulfills the job?

(d) Why do PAIRED-based algorithms seem to have a larger variance in the acceptance rate?


**Summary Of The Paper:**

This paper aims to improve a known application of curriculum generation with two new techniques. One is to select tasks according to the difficulties to the student. The other one is to reward the teacher according to the student’s gradient norm such that the teacher can generate tasks which enhance the learner more. The authors add these two techniques to existing algorithms, PAIRED and Goal GAN and experimentally test the performances with these two techniques.


**Summary Of The Review:**

This paper is suggested to be rejected because (1) the algorithm does not propose new measurements on if a task is in the zone of proximal development and does not develop novel techniques to solve the problem, and (2) the paper needs more clarification.

---

> ### Author Response · Authors · 2022-11-11
> **Thank you for the feedback! We have revised the paper to address the reviewer's concerns and ran additional experiments suggested by the reviewer.**
>
> We thank the reviewer for their thorough feedback and suggestions. We are glad that the reviewer finds our method intuitive and our experiments well-designed. We’ve revised the paper to add clarifications, new ablation studies for GRAD based on the reviewer’s suggestions, and elaboration to our motivation for ZONE. Our revisions are marked in orange.
> Below we go through the comments pointwise.
>
> > (1) This paper does not bring in enough novelty. The problem is controlling the difficulty of tasks in the zone of proximal development is already considered in the two baselines. [...] POET already proposes to reject or delete new environments that are too hard or too easy for the agent.
>
> Thanks for raising this! We agree with the reviewer (and acknowledge this in our paper) that prior works in curriculum learning have used task difficulty as a heuristic/reward for training the teacher. We’re primarily interested in formalizing the role of the teacher in aiding the student’ generalization. To that end, we’d like to draw out three differences between our work and prior work, which we’ve revised the paper to reflect.
>
> **The first difference is that we develop a novel technique GRAD**, which instantiates our notion of learning progression and has not been used in the curriculum learning literature to the best of our knowledge.
>
> **The second difference is that our work offers a unified analysis under ZONE** that formalizes why it makes sense to use difficulty combined with a notion of learning progression (Bayesian approach in Section 3 and formal RL analysis in Appendix B) **and empirically shows that ZONE can be useful for different algorithms in the literature.** This is not something that POET [1] or MCC [2] (a predecessor to POET) or other non-evolutionary curriculum learning algorithms have done. We have updated the related work according to your suggestions to point out the similarities and differences with MCC and POET.
>
> **The third difference between our work and prior (non-evolutionary) curriculum learning works is that the REJECT technique creates a decision rule for the teacher in training the student.** For example, as the reviewer notes, PAIRED does already train the teacher on task difficulty—however, the teacher acting on this decision rule (PAIRED+REJECT) is key to the significant gains we see in Figure 3!
> Albeit simple, REJECT does highlight the importance of targeting tasks that lie within the student’s ZPD without the additional cost of re-generating new tasks. We believe that this is an insight that is empirically demonstrated on hard task generation setting, and that we formalize in Section 3 and Appendix B. For regret-based curriculum methods like PAIRED, REJECT helps the student improve because the teacher targets tasks where the student still needs to learn an optimal sequence of actions. REJECT helps the teacher improve because the teacher directly trains on tasks that incur positive regret. Tasks that are rejected are ones with negative/zero regret: These are tasks where the student does better than (or equal to) the anti-student. In sparse reward settings, these tasks do not present large learning opportunities for the student. Therefore, if the teacher only keeps tasks with positive regret, it curates tasks for the student where the student has a larger learning opportunity. Directly training the teacher on the accepted problems results in the teacher learning to generate better tasks much faster (rf. Figure 5b, rejection rates).
>
> [1] Wang, Rui, et al. "Paired open-ended trailblazer (poet): Endlessly generating increasingly complex and diverse learning environments and their solutions." arXiv preprint arXiv:1901.01753 (2019).
> [2] Brant, Jonathan C., and Kenneth O. Stanley. "Minimal criterion coevolution: a new approach to open-ended search." Proceedings of the Genetic and Evolutionary Computation Conference. 2017.
>
> > (2) The motivation of the approach needs more clarification. Neither of the two proposed techniques is proven to maximize the derived objective.
>
> Thanks for raising these points! We admit that it is hard to formally prove that these techniques strictly lead to better generalization performance: We cast REJECT and GRAD as simple techniques that are easily computable, though at the cost of being less formal. However, Appendix B makes a different tradeoff: It formalizes both techniques—REJECT as minimizing the trajectory distribution between the optimal policy and the student policy, and GRAD as measuring learning potential—but at the cost of being hard to compute (eg. it assumes that we have access to the optimal policy and that we can easily compute the deviation in trajectory distribution in the function approximation setting). We are open to the reviewer’s suggestions on ways around this tradeoff of computability and formalism!

---

> > ### Author Response · Authors · 2022-11-11
> > **Continuation of revisions**
> >
> > > (2) [M]aximizing the student's gradient norm is unrelated to this main idea of generating ZPD tasks.
> >
> > Research on the zone of proximal development  in psychology, cognitive science, and education centers on the notion of how a student can most quickly learn from an expert/experienced adult [3,4,5]. Thus, our work focuses on directly accelerating the student’s learning progress and generating the largest learning gains in the student, which is the central objective of tuning the ZPD. This is why maximizing the student’s gradient norm is tied to ZPD. GRAD is a novel technique in the curriculum learning literature: Prior curriculum learning works only focus on task difficulty, however without explicitly controlling or incentivizing for accelerated student learning.
> >
> > [3] Tennie, Claudio, Josep Call, and Michael Tomasello. "Ratcheting up the ratchet: on the evolution of cumulative culture." Philosophical Transactions of the Royal Society B: Biological Sciences 364.1528 (2009): 2405-2415.
> >
> > [4] Shafto, Patrick, Noah D. Goodman, and Thomas L. Griffiths. "A rational account of pedagogical reasoning: Teaching by, and learning from, examples." Cognitive psychology 71 (2014): 55-89.
> >
> > [5] Wass, Rob, and Clinton Golding. "Sharpening a tool for teaching: the zone of proximal development." Teaching in Higher Education 19.6 (2014): 671-684.
> >
> >
> > > (3) More description of the setting is needed. It is unclear whether the paper focuses on a generalization objective to maximize expected returns among a given environment distribution or what.
> >
> > We’ve updated the paper to clarify that the generalization objective (Equation 1) measures the expected returns over a held-out task distribution ($c^{\text{test}} \sim \chi(C^{\text{test}})$. The objective maximizes over the teacher’s distribution of tasks, which the teacher can control.
> >
> > > (4) Modelling [REJECT as] the probability in the objective in section 3.3 on page 5 [p(rcurr|ccurr,θcurr)] is unclear and technically incorrect. [...] Furthermore, this probability is decided by the dynamics of the MDP and cannot be influenced by the teacher.
> >
> > Thanks for bringing this up! Our rationale for modeling REJECT is elaborated in the paragraph preceding Section 3.3, and we’ve updated the paper to make it more clear: The teacher influences the difficulty of the curriculum by picking $c_{\text{curr}}$ and simultaneously the student’s learning progression. We note the following cases:
> >
> > - Easy tasks (high ${\color{red}{p(R^{\text{curr}}=1| c^{\text{curr}}, {\theta}^{\text{curr}})}}$) do not change the student's current model; if the student does not have a good model ($\theta^{\text{curr}} \neq \theta^*$), then picking an easy task will not increase the chances of the student learning a better test model (low ${\color{blue}{p( {\theta}^{*}|r^{\text{curr}}, c^{\text{curr}}, {\theta}^{\text{curr}})}}$).
> >
> > - The teacher should avoid picking hard tasks too: the probability of failure is high, and the student will likely need prolonged interaction with difficult task instances before learning a good test model.
> >
> > Thus, REJECT targets tasks that are neither too easy nor too hard via measures of task difficulty $p(r_{\text{curr}}|c_{\text{curr}},\theta_{\text{curr}})$. As mentioned in a previous response to the reviewer, we’ve cast REJECT and GRAD as simple techniques that are easily computable; a formal notion of REJECT that the reviewer might find more faithful to the probability is given in Appendix B, however it is hard to compute and use (eg. it assumes that we have access to the optimal policy and that we can easily compute the deviation in trajectory distribution in the function approximation setting).
> >
> > The probability $p(r_{\text{curr}}|c_{\text{curr}},\theta_{\text{curr}})$ does condition on the task parameters $c_{\text{curr}}$, as the reward is dependent on the MDP dynamics. Note that the teacher has influence over the choice of task, ie. how the teacher parameterizes $\chi_{\phi}$. By picking certain tasks over others, it chooses to train the student in different MDPs, therefore influencing the probability of success for the student.
> >
> >
> > > (5) The test on the gradient maximization idea is limited. Some ablation studies to test the gradient maximization idea alone would be interesting.
> >
> > Thanks for this suggestion! We agree with the reviewer that this is an interesting study to run. We’ve included the result of only GRAD over 4 seeds in Appendix G, Figure 9. GRAD shows competitive performance against PAIRED across the four Minigrid environments. This suggests that directing the curriculum to target the student’s learning progression (ie. maximize its gradient norm) might be a promising avenue for future curriculum learning algorithms in hard task domains.

---

> > > ### Author Response · Authors · 2022-11-11
> > > **Continuation of revisions**
> > >
> > > > (6) Some descriptions of experiments are missing. For example, how to set the gradient norm and returns scale when applying the gradient maximization technique? what contexts are all algorithms tested on for tasks (a)-(g) in Figure 2 on page 7? How the rejection rate is computed in Figure 5 on page 9?
> > >
> > > Thanks for raising these concerns! We’ve updated the paper to address these questions, and provide the answers here as well:
> > >
> > > - How to set norms and returns scale: We only experimented with adding and multiplying the returns with the norms. We note under “Modelling … norms (GRAD)” in Section 3 that in dense reward settings, the gradient norm is much larger than the success reward, therefore we multiply the two in this setting.
> > > - What contexts in Figure 2: The reported performance in Figure 2 shows the student’s performance on held-out contexts/tasks, as we care about generalization outside of the student’s training distribution.
> > > - Computation of rejection rate in Fig. 5: The teacher samples a batch of size |B| tasks. We evaluate the student as detailed in Section 3.3 to calculate $x$, the number of tasks which do not satisfy PAIRED’s regret objective (aka. tasks that have non-positive regret). We report $\frac{x}{|B|}$ as the rejection rate.
> > >
> > >
> > > > (7a) Why are there so many tasks given by the PAIRED rejected? Does PAIRED generate many negative regret tasks?
> > >
> > > We currently reject tasks where $\text{regret} <= 0$. This rejects tasks that both have negative regret and zero regret. Zero regret occurs when the student and anti-student both do not succeed on the task. At the beginning of training, we expect 50% of the tasks to have negative regret (50% chance for either student to do better), and the rest to come from the zero regret tasks.
> > >
> > > > (7b) It is shown in Table 1 on page 9 that PAIRED+REJECT+GRAD gives hard tasks from the beginning. Does it break the increasing difficulty set by the curriculum learning? Is it good?
> > >
> > > Thanks for raising this point! **The reason the environments look harder is that student in PAIRED+REJECT+GRAD has made progress in less training time.** They are designed for a more competent student. We’ve added to Appendix F earlier screenshots of the environments, which show that PAIRED+REJECT+GRAD does generate easy tasks at the start, but is able to much more quickly move to harder environments. This contributes to the student learning much faster/better with PAIRED+REJECT+GRAD than with PAIRED.
> > >
> > > > (7c) REJECT does not improve Goal GAN much. Can we say PAIRED is not controlling the difficulty of tasks well as claimed, but Goal GAN already fulfills the job?
> > >
> > > Yes! We believe this is the case that naive PAIRED takes much longer to learn to control the difficulty of the task. Under experiments Section 5.1, we note that the PAIRED teacher has a much harder learning problem than the Goal GAN teacher: the PAIRED teacher has to generate an entire $15 \times 15$ grid with up to $50$ walls, whereas the GoalGAN teacher generates only a 2D goal. It is much harder for the PAIRED teacher to control for difficulty as small variations in its generations can result in large variations of the task difficulty. For example, placing a single wall piece at the end of a corridor might block off a direct route to the goal for the student. By contrast, in the Goal GAN/control environments which typically have little to no obstacles, small variations the 2D goals would not make a large difference in the task difficulty.
> > >
> > > > (7d) Why do PAIRED-based algorithms seem to have a larger variance in the acceptance rate?
> > >
> > > We hypothesize that this is because of the aforementioned learning challenge for the teacher: It is much harder for the teacher to learn how to generate ZPD tasks. Usually when the teacher does get positive reward, it is able to continue generating ZPD tasks, however this happens by chance and it is not a consistent finding. We believe this causes the large variance in acceptance/rejection rates.

---

### Official Review · Reviewer_Go3r · 2022-10-25

**Confidence:** 5
**Correctness:** 3
**Technical Novelty And Significance:** 3
**Empirical Novelty And Significance:** 2
**Recommendation:** 5

**Clarity, Quality, Novelty And Reproducibility:**

The paper is good in clarity and novelty, and seems easy to reproducibility. However, the experiment part is not sufficient and there is still room for improvement in experimental analysis.

**Strength And Weaknesses:**

Strengths:
1.	The paper is well written and easy to understand.
2.	The motivation of the method is intuitive and backed by psychology.
Weaknesses:
1.	The transition from psychological concepts to computational metrics seems not that natural, which causes some problems. For instance, the difficulty needs manual design based on different RL models, i.e., a fixed range for Goal GAN and a changeable range for PAIRED. Besides, the GRAD technique cannot be applied when the teacher is not trained with rewards.
2.	The number of baselines is a bit small, which degrades its universality and generality.
3.	The analysis of the experiment is not convincing. Figure 5a shows that ZONE generates more complex tasks, instead of better tasks matching the learning situation of a student. The more obstacle-filled or hallways-filled tasks are not necessarily better. Figure 5c shows that the gradient norm of student model without GRAD is larger than that with GRAD in the later stage of training, which seems weird and strange.


**Summary Of The Paper:**

The paper proposes ZONE, a computational framework of curriculum task generation for reinforcement learning based on a concept from developmental psychology, i.e., zone of proximal development (ZPD). The target of the framework is to measure and schedule difficulty and progression of a student model. The approach is composed of two techniques, REJECT and GRAD. REJECT is to replace too easy or too hard tasks generated by teacher with proper tasks. GRAD is to maximize the norm of student model’s gradient, which reflects student progression. To evaluate the effectiveness, the authors conduct experiments on two baselines and their corresponding settings.

**Summary Of The Review:**

To summarize, the paper is good except for the experiments part. The motivation is interesting, based on developmental psychology. The method is well described and easy to understand. However, the experimental results are not so ideal and the analysis is not so convincing.

---

> ### Author Response · Authors · 2022-11-11
> **Thank you for your feedback! We have revised the paper to address the reviewer's concerns.**
>
> We thank the reviewer for their feedback and suggestions. We’ve updated our paper to reflect the following changes in light of their concerns. Our revisions are marked in orange.
>
> > The transition from psychological concepts to computational metrics seems not that natural, which causes some problems. For instance, the difficulty needs manual design based on different RL models, i.e., a fixed range for Goal GAN and a changeable range for PAIRED. Besides, the GRAD technique cannot be applied when the teacher is not trained with rewards.
>
> We’d like to point out that the goal of our work is to offer a formalism (ZONE) as a lens into the role of the teacher (Section 3 and Appendix B), and a simple toolbox for improving existing curriculum generation algorithms. The goal of our work is _not_ to innovate on better metrics for capturing difficulty.  Determining “right” choice of difficulty is out of scope of our work, however we agree that it would be important for future work to understand!
>
> We admit that the GRAD technique is limited to teachers that are trained with rewards, however prior works suggest that reward-trained or reward-sensitive teachers are a promising way for doing curriculum learning both for RL agents and real human students [1,2,3,4].
>
> [1] Yuqing Du, Pieter Abbeel, and Aditya Grover. It takes four to tango: Multiagent self play for automatic curriculum generation. In International Conference on Learning Representations, 2022
>
> [2] Dennis, Michael, et al. "Emergent complexity and zero-shot transfer via unsupervised environment design." Advances in neural information processing systems 33 (2020): 13049-13061.
>
> [3] Bassen, Jonathan, et al. "Reinforcement learning for the adaptive scheduling of educational activities." Proceedings of the 2020 CHI Conference on Human Factors in Computing Systems. 2020.
>
> [4] Rafferty, Anna N., et al. "Faster teaching via pomdp planning." Cognitive science 40.6 (2016): 1290-1332.
>
> > 2. The number of baselines is a bit small, which degrades its universality and generality.
>
> We are open to the reviewer’s suggestions on improving our evaluations.
>
> > 3. The analysis of the experiment is not convincing. Figure 5a shows that ZONE generates more complex tasks, instead of better tasks matching the learning situation of a student. The more obstacle-filled or hallways-filled tasks are not necessarily better.
>
> Thanks for raising this point! **The reason the environments look harder is that student in PAIRED+REJECT+GRAD has made progress in less training time.** They are designed for a more competent student. We’ve added to Appendix F earlier screenshots of the environments, which show that PAIRED+REJECT+GRAD does generate easy tasks at the start, but is able to much more quickly move to harder environments. This leads to the environments in 5a also being more challenging and better curated.
>
> We are open to other suggestions that the reviewer might have to improve the quality of the analysis.
>
> > Figure 5c shows that the gradient norm of student model without GRAD is larger than that with GRAD in the later stage of training, which seems weird and strange.
>
> We’d like to note that the variance in PAIRED gradient norm is large around 1.5x10^8 steps, and is not significantly larger than any of the other curves including PAIRED+GRAD.

---

### Decision · Program_Chairs · 2023-01-20

**Decision:**

Reject

**Justification For Why Not Higher Score:**

The paper had the following significant deficiencies: lack of clarity in terms of contribution and motivation as well as clear performance advantage and relevant baselines.

**Justification For Why Not Lower Score:**

N/A

**Metareview: Summary, Strengths And Weaknesses:**

The authors proposed a framework for curriculum task generation in reinforcement learning called Zone, where a “teacher” network generates tasks to enable “student” learning. The framework adds two techniques to existing algorithms. The REJECT technique removes tasks based on the learner's difficulty, and the GRAD technique rewards the teacher according to the student’s gradient norm such that the teacher can generate tasks that enhance the learner more.

Reviewers appreciated the work for being easy to understand and intuitive. However, they pointed out the issues of weak experimental results and the lack of clarity on the novelty and the motivation of the contribution.

I met with the reviewers to discuss these issues. Based on the discussion, we agreed that the paper has the following significant deficiencies that remain unresolved:
- The lack of performance improvement compared to goal-gan as well as the lack of comparisons with highly relevant baselines such as amigo indicates further empirical investigation and room for improvement.
- Although the work focuses on difficulty control, GRAD as a mechanism for accelerating learning is proposed as a significant contribution. The connection between them is unclear.
- Unification is mentioned as a novelty and contribution, but the scope of unification remains specific to a narrow class of mechanisms.
- These last two points indicate that the main message about the motivation and contribution of the work could not be fully appreciated, which can be addressed by focusing on these parts for the next version of the paper.

Addressing these deficiencies requires a significant revision of the current draft. Still, we believe that a resubmission that carefully addresses these concerns will make this paper much stronger and above a borderline paper.


**Summary Of Ac-Reviewer Meeting:**

I met with the reviewers to discuss these issues. Based on the discussion, we agreed that the paper has the following significant deficiencies that remain unresolved:
The lack of performance improvement compared to goal-gan as well as the lack of comparisons with highly relevant baselines such as amigo indicates further empirical investigation and room for improvement.
Although the work focuses on difficulty control, GRAD as a mechanism for accelerating learning is proposed as a significant contribution. The connection between them is unclear.
Unification is mentioned as a novelty and contribution, but the scope of unification remains specific to a narrow class of mechanisms.
These last two points indicate that the main message about the motivation and contribution of the work could not be fully appreciated, which can be addressed by focusing on these parts for the next version of the paper.